# Fair Clustering Under a Bounded Cost

**Seyed Esmaeili**
University of Maryland
esmaeili@cs.umd.edu

**Brian Brubach**
Wellesley College
bb100@wellesley.edu

**Aravind Srinivasan**
University of Maryland
asriniv1@umd.edu

**John P. Dickerson**
University of Maryland
johnd@umd.edu

## Abstract

Clustering is a fundamental unsupervised learning problem where a dataset is partitioned into clusters that consist of nearby points in a metric space. A recent variant, *fair* clustering, associates a *color* with each point representing its group membership and requires that each color has (approximately) equal representation in each cluster to satisfy group fairness. In this model, the cost of the clustering objective increases due to enforcing fairness in the algorithm. The relative increase in the cost, the "price of fairness," can indeed be unbounded. Therefore, in this paper we propose to treat an upper bound on the clustering objective as a constraint on the clustering problem, and to maximize equality of representation subject to it. We consider two fairness objectives: the group utilitarian objective and the group egalitarian objective, as well as the group leximin objective which generalizes the group egalitarian objective. We derive fundamental lower bounds on the approximation of the utilitarian and egalitarian objectives and introduce algorithms with provable guarantees for them. For the leximin objective we introduce an effective heuristic algorithm. We further derive impossibility results for other natural fairness objectives. We conclude with experimental results on real-world datasets that demonstrate the validity of our algorithms.

## 1  Introduction

Machine learning algorithms are increasingly being applied to settings that directly influence human lives. This has spurred a growing *fair machine learning* community [9], which develops machine learning algorithms that are made to satisfy certain fairness criteria. Choosing an appropriate definition of fairness—and even deciding *if* explicitly defining fairness is appropriate to begin with—is a morally-laden and application-specific decision [28, 43]. We make no normative statements here; rather, we focus on a commonly-used and often legally-backed family of fairness definitions—*group fairness*—in the context of *clustering*, arguably the most fundamental unsupervised learning problem.

A recent group-membership fairness definition, called *fair clustering* in the literature, has received significant interest [20, 10, 11, 2, 7, 31, 22, 8, 3, 33, 1]. In fair clustering, each point has a color that designates its group membership, and a clustering objective such as $k$-median or $k$-means is given. The goal is to find a clustering that minimizes the objective subject to the constraint that each cluster has each color represented within some pre-specified proportions. For example, there may be two colors, red and blue, and the constraint could require per-color representation between 40% and 60%.

An acknowledged fact in fair clustering—and, indeed, in many allocation and matching settings—is that the fairness (e.g., proportion) constraint could cause degradation in the clustering objective [12, 19]. A point may be assigned to a further away center (cluster) to satisfy the proportion constraint [20].

35th Conference on Neural Information Processing Systems (NeurIPS 2021).

The degradation in the objective due to the imposed fairness constraint is called the *price of fairness* (PoF), mathematically defined as PoF = (*cost of fair solution*) / (*cost of agnostic solution*).

Unlike some examples in the literature [12, 21], the price of fairness in the case of fair clustering is unbounded, as seen in Figure 1. By enforcing a form of group fairness requiring an even split across colors in each cluster, a fair clustering algorithm would perform arbitrarily poorly as the two groups of points separate in space, while a "color-blind" algorithm would remain unchanged (bottom-left and bottom-right of Figure 1, respectively). The possibly unbounded increase in the clustering cost (unbounded price of fairness) indicates that fair clustering can yield clusters consisting of points that are far apart in the metric space instead of combining nearby points—often the main motivation behind clustering in machine learning and data

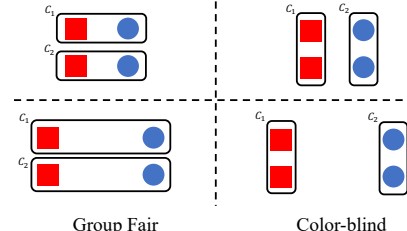

Figure 1: *Comparison between group fair (left) and color-blind (right) clustering. Unlike color-blind clusters, group fair clusters may combine faraway points (bottom-left).*

analysis. Furthermore, the legal notion of disparate impact does not force an organization to output a fair clustering if it can justify an unfair one due to "business necessity," i.e., potential loss in quality [45, 44]. This possible conflict between the clustering objective and the fairness constraint indicates the need for fair clustering algorithms that operate in a setting where the clustering cost cannot exceed a pre-set upper bound.

**Our Contributions.** In this paper, we address fair clustering under an exogenous threshold on the clustering objective. We formulate the problem mathematically in a general setting that captures all of the traditional $k$-clustering objectives, i.e. $k$-center, $k$-median, and $k$-means. Throughout, we focus on two general formulations of group fairness: GROUP-UTILITARIAN and GROUP-EGALITARIAN, along with the GROUP-LEXIMIN objective which generalizes the traditional GROUP-EGALITARIAN definition. We show that these objectives lead to problems that are NP-hard in general. Further, assuming $P \neq NP$ we derive lower bounds on the additive approximation of any polynomial time algorithm for the GROUP-UTILITARIAN and GROUP-EGALITARIAN objectives. We provide bi-criteria approximation algorithms for the GROUP-UTILITARIAN and GROUP-EGALITARIAN objectives in which the constraint has a bounded violation and the objective is bounded from the optimal value by an additive error. For the GROUP-LEXIMIN objective we provide an effective heuristic. Further, we consider other possibly more "flexible" fairness objectives, but demonstrate inapproximability results for them. Finally, we test the performance of our algorithms on a collection of datasets and see that we obtain good solutions with low "fairness violations." We note that due to the page limit, all proofs are placed in Appendix A.

## 2 Related Work

The metric clustering problems $k$-center, $k$-median, and $k$-means are fundamental in unsupervised learning and operations research. All are NP-hard with a long line of research on approximation algorithms. For $k$-center, two distinct algorithms achieve a 2-approximation which is tight assuming $P \neq NP$ [26, 23, 27]. The current best approximation for $k$-median is a $(2.675 + \epsilon)$-approximation in $n^{O((1/\epsilon)\log(1/\epsilon))}$ time [18], and for $k$-means, there is a 6.357-approximation [4].

The *fair* clustering problem with group fairness constraints was proposed by [20]. They studied $k$-center and $k$-median in a setting with only two colors. Followup work by [11, 10, 7, 31] gave extensions to the $k$-means objective, more than two colors, multi-color points (i.e., intersecting demographic groups), and scalability. Other works by [38, 15, 16] look at non-group-fairness definitions; the former investigates individual fairness, while the latter two address probabilistic fairness guarantees for pairs or communities of points. The aforementioned works optimize the clustering objective subject to fairness constraints; however, satisfying the fairness constraints may come at the expense of a significant increase in the clustering objective. Accordingly [48] and very recently [37] explored the cost/fairness tradeoff, but using a multi-objective approach. Unlike our work they do not establish approximation guarantees. Further, the fairness objectives used are different from ours, i.e. in [48] the fairness objective is penalized for proportions that are not precisely equal to the population level while in [37] it is penalized for color ratios that are not equal to 1. Moreover, in [48] the cost/fairness tradeoff is a non-monotone function of a parameter which the user must adjust, while [37] only focuses on the $k$-means objective and provides convergence guarantees only for a smoothed version of the original problem.

We focus on the formal tradeoff between fairness and the clustering objective. We note that the clustering objective can be replaced by the price of fairness (PoF). However, in our setting, the results are clearer if we refer to the clustering objective instead of the PoF. Ultimately a higher PoF corresponds to a weakly higher fair clustering objective and vice versa. In fact, they are multiples of one another: PoF = (*cost of fair solution*) / (*cost of agnostic solution*). This tradeoff between fairness and efficiency manifested the PoF concept, in operations research by [12] and simultaneously in computer science by [19], showing general approaches to defining and measuring it. Similar to our work, others have adapted PoF as a hard constraint in emergency response [30], organ allocation [39], and rideshare [36, 42], and a partial constraint in scarce resource allocation for kidney dialysis [29] and organ exchange [40]. We propose a framework for balancing PoF for a traditional "efficient" objective in clustering, which finds application in areas such as advertising, network analysis, and data summarization.

## 3 Preliminaries

In a clustering problem, we are given a set of points $\mathcal{C}$ in a metric space. A distance function $d(i,j)$ specifies the distance between each pair of points $i, j \in \mathcal{C}$. Furthermore, $d$ is symmetric, non-negative, and satisfies the triangle inequality. Common clustering cost (loss) functions (e.g., $k$-means or $k$-median) can be written as $\min_{S:|S|\leq k,\phi} L_p^k(\mathcal{C}) = \min_{S:|S|\leq k,\phi} \left( \sum_{j\in\mathcal{C}} d^p(j,\phi(j)) \right)^{1/p}$ where $k$ is the number of clusters, $S$ is the set of cluster centers chosen from a candidate set of centers $\mathcal{S}$, and $\phi\colon \mathcal{C} \to S$ is an assignment function that assigns points to cluster centers. The value $p$ determines the type of clustering, i.e., $p = \infty, 1$, and $2$ for $k$-center, $k$-median, and $k$-means, respectively.

In fair clustering, each point has a color associated with it to indicate its group membership. Specifically, we have a function $\chi\colon \mathcal{C} \to \mathcal{H}$ where $\mathcal{H}$ is the set of possible colors. We denote the set of all points of color $h$ by $\mathcal{C}^h$. The fair clustering problem (**FC**) [20, 10, 11, 2, 7, 31, 8] is to minimize the clustering objective while satisfying additional fairness constraints:

$$\min_{S:|S|\leq k,\phi} \left( \sum_{j\in\mathcal{C}} d^p(j,\phi(j)) \right)^{1/p} \tag{1a}$$

$$\text{s.t. } \forall i \in S, \forall h \in \mathcal{H} : \beta_h|\mathcal{C}_i| \leq |\mathcal{C}_i^h| \leq \alpha_h|\mathcal{C}_i| \tag{1b}$$

where $\mathcal{C}_i$ is the set of points in cluster $i$ and $\mathcal{C}_i^h \subseteq \mathcal{C}_i$ is the subset of points in cluster $i$ with color $h$. $\beta_h$ and $\alpha_h$ are pre-specified lower and upper proportion bounds for color $h$, respectively. Clearly, $0 < \beta_h \leq \alpha_h < 1$.

In "unfair" clustering problems, the assignment function $\phi$ maps points to the nearest center in $S$, i.e., $\phi(j) = \text{argmin}_{i\in S} d(i,j)$ since this minimizes the objective. However, satisfying the added constraints in fair clustering may cause points to be assigned to clusters that are farther away. This motivates the fair assignment problem (**FA**), in which the set of centers $S$ is given and the objective is to minimize the clustering cost subject to fairness constraints:

$$\min_{\phi} \left( \sum_{j\in\mathcal{C}} d^p(j,\phi(j)) \right)^{1/p} \tag{2a}$$

$$\text{s.t. } \forall i \in S, \forall h \in \mathcal{H} : \beta_h|\mathcal{C}_i| \leq |\mathcal{C}_i^h| \leq \alpha_h|\mathcal{C}_i| \tag{2b}$$

The only difference between the fair assignment (2) and fair clustering (1) problems is that $S$ is not an optimization variable in the fair assignment problem.

## 4 Fair Clustering Under a Bounded Cost (FCBC)

The fundamental idea of fair clustering under a bounded cost (**FCBC**) is to minimize a measure of unfairness subject to an upper bound on the clustering cost:

$$\min \text{ Unfairness} \tag{3a}$$

$$\text{s.t. Clustering Cost} \leq \text{Given upper bound} \tag{3b}$$

Next, we transform (3a) and (3b) above into a clear mathematical optimization problem.
**The Constraint (3b):** The clustering cost is $\left( \sum_{j\in\mathcal{C}} d^p(j,\phi(j)) \right)^{1/p}$. Let $U$ denote the exogenous upper bound on clustering cost. Then, (3b) becomes $\left( \sum_{j\in\mathcal{C}} d^p(j,\phi(j)) \right)^{1/p} \leq U$. Note that for the case of the $k$-center where $p = \infty$, the constraint reduces to a simpler form, specifically $\forall j \in \mathcal{C}, d(j,\phi(j)) \leq U$.

**The Objective (3a):** In prior work, a given clustering is considered fair if for each cluster, the proportions of each color lie within pre-specified lower and upper bounds, i.e.: $\forall i \in S, \forall h \in \mathcal{H}$ : $\beta_h |\mathcal{C}_i| \le |\mathcal{C}_i^h| \le \alpha_h |\mathcal{C}_i|$. However, bounding the clustering cost may make it impossible to have a fair feasible solution. Therefore, we instead set a measure of unfairness for each color and try to minimize this measure. Let $\Delta_h$ denote the worst proportional violation across the clusters for a color $h$. Specifically, for a given clustering, $\Delta_h \in [0, 1]$ is the minimum non-negative value such that:

$$\forall i \in S : (\beta_h - \Delta_h)|\mathcal{C}_i| \le |\mathcal{C}_i^h| \le (\alpha_h + \Delta_h)|\mathcal{C}_i|. \tag{4}$$

Clearly, if $\Delta_h = 0$, then color $h$ is within the desired proportion in every cluster. Having set $\Delta_h$ to be a measure of the unfair treatment that group $h$ receives, we are faced with the question of setting the fairness objective, for which there are many reasonable options. We consider two prominent and intuitive fairness objectives [14]:

$$\text{GROUP-UTILITARIAN} = \min \sum_{h \in \mathcal{H}} \Delta_h \quad , \quad \text{GROUP-EGALITARIAN} = \min \max_{h \in \mathcal{H}} \Delta_h$$

The GROUP-UTILITARIAN objective minimizes the sum of proportional violations for all of the colors, treating all points of a specific color as a single player in a game. The GROUP-EGALITARIAN objective minimizes the maximum proportional violation across the colors. We also consider a more generalized version of the GROUP-EGALITARIAN objective, the GROUP-LEXIMIN objective. Like GROUP-EGALITARIAN, the GROUP-LEXIMIN objective minimizes the maximum (worst) violation, but it goes further to minimizes the second-worst violation, then the third-worst violation, and so on until no further improvement can be made. We now state the fair clustering under a bounded cost problem (**FCBC**):

$$\min_{S:|S| \le k, \phi} \text{UNFAIRNESS-OBJECTIVE} \tag{5a}$$
$$\text{s.t.} \quad \Big( \sum_{j \in \mathcal{C}} d^p(j, \phi(j)) \Big)^{1/p} \le U \tag{5b}$$

where the UNFAIRNESS-OBJECTIVE could equal GROUP-UTILITARIAN, GROUP-EGALITARIAN, or GROUP-LEXIMIN. Similar to the fair assignment **FA** problem (2), we may define the fair assignment under a bounded cost (**FABC**) problem as:

$$\min_{\phi} \text{UNFAIRNESS-OBJECTIVE} \tag{6a}$$
$$\text{s.t.} \quad \Big( \sum_{j \in \mathcal{C}} d^p(j, \phi(j)) \Big)^{1/p} \le U \tag{6b}$$

where similarly the optimization is over the assignment function $\phi$ while the set of centers $S$ is fixed.

## 5 Hardness of FCBC & FABC

First, we formally state the hardness of the fair clustering **FC** and the fair assignment **FA** problems.

**Theorem 5.1.** *The fair clustering* **FC** *(1) and fair assignment* **FA** *(2) problems are NP-hard.*

We now establish the hardness of fair clustering under a bounded cost **FCBC** and fair assignment under a bounded cost **FABC**. We note that these hardness results follow for all objectives (GROUP-UTILITARIAN, GROUP-EGALITARIAN, and GROUP-LEXIMIN).

**Theorem 5.2.** *Fair clustering under a bounded cost* **FCBC** *and fair assignment under a bounded cost* **FABC** *are NP-hard.*

Although we have shown that both the fair clustering and fair assignment problems under a bounded cost are NP-hard, the reductions rely on setting the upper bound $U$ to a small enough value, precisely that of the optimal fair clustering cost. It seems reasonable to expect both problems to transition into being polynomial time solvable if the upper bound becomes sufficiently large. We show in Section 8 that such a result is not easy to establish and would lead to a true approximation for fair clustering which is yet to be produced in the fair clustering literature for arbitrary metric spaces and arbitrary lower and upper color proportion bounds.

For a given clustering cost $U$, there are many clusterings (solutions) of cost not exceeding $U$. Let $\mathcal{S}_U$ be the set of those solutions, i.e. if $(S_t, \phi_t) \in \mathcal{S}_U$, then $(S_t, \phi_t)$ is a clustering with a cost that does not exceed $U$. Further, let $L_t$ be the size of the smallest non-empty cluster[1] in the clustering $(S_t, \phi_t)$,

---

[1] An empty cluster is a cluster with no points assigned to it. This could happen if for example the assignment function $\phi$ does not map any point to a a given center including the center itself.

then we define $L(U)$ to be the size of the smallest cluster across all clusterings of cost not exceeding $U$, i.e. $L(U) = \min_{(S_t, \phi_t) \in \mathcal{S}_U} L_t$. Clearly, for $U_1$ and $U_2$ such that $U_2 \geq U_1$, then $L(U_2) \leq L(U_1)$ since $\mathcal{S}_{U_1} \subseteq \mathcal{S}_{U_2}$. We can conclude the following fact from the definition of $L(U)$:

**Fact 5.1.** *For a given upper bound $U$, no clustering with cost less than or equal to $U$ can have less than $L(U)$ many points in a non-empty cluster.*

We show that the quantity $L(U)$ plays a fundamental role. In fact, lower bounds on the additive approximation[2] for the proportional violations and fairness objectives are related to $L(U)$ as shown in the following theorem:

**Theorem 5.3.** *For a given instance of the* **FCBC** *or* **FABC** *problem with an arbitrary upper bound $U$, unless $P = NP$ no polynomial time algorithm can produce a solution with a cost not exceeding $U$ that satisfies any of the following conditions: (a) The proportional violation of any color $h \in \mathcal{H}$ is $\Delta_h < \frac{1}{8L(U)}$. (b) The additive approximation for the* GROUP-UTILITARIAN *objective is less than $\frac{|\mathcal{H}|}{8L(U)}$. (c) The additive approximation for the* GROUP-EGALITARIAN *objective is less than $\frac{1}{8L(U)}$.*

# 6 Algorithms for FCBC

Our main result for the **FCBC** problem is the following theorem which follows as a direct consequence of the guarantees of Theorems 6.2, 6.4, 6.5, 6.6, and 6.7:

**Theorem 6.1.** *For any clustering objective, given a bound $U$ on the clustering cost, Algorithm 1 solves the fair clustering under a bounded cost* **FCBC** *problem at a cost of at most $U' = (2 + \alpha)U$ where $\alpha$ is the approximation ratio of the color-blind clustering algorithm. The additive approximation is $|\mathcal{H}|(\epsilon + \frac{2}{L(U')})$ for the* GROUP-UTILITARIAN *objective and $\epsilon + \frac{2}{L(U')}$ for the* GROUP-EGALITARIAN *objective.*

From the theorem above, it is clear that the additive approximation guarantees we have improve when the cost does not permit small clusters. Indeed, in the absence of outlier points and for reasonable values of $k$, small clusters are unlikely to exist. Further, empirically we verify the smallest cluster size and find that the smallest cluster size is 159 points (see Section 9.3). See Appendix C for more discussion.

We now provide our general algorithm for fair clustering under a bounded cost **FCBC** which we denote by **ALG**-**FCBC**($U$, UNFAIRNESS-OBJECTIVE) where we have made explicit reference to the dependence of **ALG**-**FCBC** on the given cost upper bound $U$ and the desired UNFAIRNESS-OBJECTIVE which could either be the GROUP-UTILITARIAN, GROUP-EGALITARIAN, or GROUP-LEXIMIN objective.

**ALG**-**FCBC**($U$, UNFAIRNESS-OBJECTIVE) (see Algorithm 1) involves two steps, in step **(1):** we use a color-blind approximation algorithm to find the cluster centers $S$, in step **(2):** we call the algorithm **ALG**-**FABC**($S, U'$, UNFAIRNESS-OBJECTIVE) for the **FABC** problem. It should be noted that we have fed **ALG**-**FABC** the set of centers $S$ from step **(1)**, further the cost upper bound for **ALG**-**FABC** is set to $U' = (2 + \alpha)U$ while the UNFAIRNESS-OBJECTIVE remains unchanged. We further note that **ALG**-**FABC** will have the same clustering objective as **ALG**-**FCBC**, e.g. if **ALG**-**FCBC** is given the $k$-median objective so well **ALG**-**FABC**.

Clearly, from algorithm **ALG**-**FCBC** the **FCBC** problem is closely related to the **FABC** problem. In fact, we establish the following general theorem for all clustering objectives: $k$-center, $k$-median, and $k$-means that shows that an algorithm which solves the **FABC** problem with provable guarantees can be used to solve the **FCBC** problem with provable guarantees:

**Theorem 6.2.** *For any clustering objective and both the* GROUP-UTILITARIAN *and* GROUP-EGALITARIAN *objectives, given an algorithm that solves fair assignment under a bounded cost* **FABC** *with additive approximation $\mu$, the fair clustering under a bounded cost* **FCBC** *problem can be solved with an additive approximation of $\mu$ and at a cost of at most $(2 + \alpha)U$, where $\alpha$ is the approximation ratio of the color-blind clustering algorithm.*

---

[2]An algorithm for a minimization problem with additive approximation $\mu > 0$, returns a value for the objective that is at most OPT $+\mu$ where OPT is the optimal value.

---
**Algorithm 1** :**ALG-FCBC**($U$, UNFAIRNESS-OBJECTIVE)
---
1: Choose a set of centers $S$ by running a color-blind clustering algorithm of approximation ratio $\alpha$.
2: Set $U' = (2 + \alpha)U$ and call **ALG-FABC**($S, U'$, UNFAIRNESS-OBJECTIVE)
---

## 6.1 Fair Assignment Under a Bounded Cost

Algorithm block 2 shows the steps of our algorithm **ALG-FABC** for the **FABC** objective. In step **(1):** we search for the optimal proportional violations given the bound on the clustering cost $U$ using LPs. Having found the near-optimal solution, in step **(2):** we round the possibly fractional solution to a feasible integer solution using a netowk flow algorithm. We note that the details of the search done in step **(1)** depend on the objective, i.e., GROUP-UTILITARIAN or GROUP-EGALITARIAN.

---
**Algorithm 2** :**ALG-FABC**($S, U$, UNFAIRNESS-OBJECTIVE)
---
1: Given the UNFAIRNESS-OBJECTIVE, search for the optimal proportion violation values $\Delta_h$ at a cost upper bound of $U$ using the feasibility LPs of (7).
2: Apply network flow rounding to the LP solution with the optimal value.
3: **return** the set of centers $S$ and the assignment function $\phi$ (resulting from the rounded LP solution).
---

We note that in fair assignment under a bounded cost **FABC** the set of centers $S$ has already been chosen and the optimization is done only over the assignment $\phi$ of points to centers. We let $x_{ij}$ be a decision variable that equals 1 if point $j$ is assigned to center $i \in S$ and 0 otherwise. Note that the values of $x_{ij}$ are a way to represent the assignment function $\phi$. Regardless of the objective that is being minimized, the following set of constraints must hold:

$$\sum_{i,j} d^p(i,j)x_{ij} \leq U^p \tag{7a}$$

$$\forall j \in \mathcal{C} : \sum_{i \in S} x_{ij} = 1, \quad x_{ij} \in [0,1] \tag{7b}$$

$$\forall h \in \mathcal{H} : \Delta_h \in [0,1] \tag{7c}$$

$$\forall h \in \mathcal{H}, \forall i \in S : (\beta_h - \Delta_h)\Big(\sum_{j \in \mathcal{C}} x_{ij}\Big) \leq \sum_{j \in \mathcal{C}^h} x_{ij} \leq (\alpha_h + \Delta_h)\Big(\sum_{j \in \mathcal{C}} x_{ij}\Big) \tag{7d}$$

For the $k$-center problem, the first constraint (7a) is replaced by $\forall j \in \mathcal{C} : x_{ij} = 0$ if $d(i,j) > U$. Note that in the above we have $x_{ij} \in [0,1]$ which is a relaxation of $x_{ij} \in \{0,1\}$, as the latter would result in an intractable mixed-integer program. With our variables being $x_{ij}$ and $\Delta_h$ it is reasonable to consider a convex optimization approach. That is, we could choose to minimize the objective GROUP-UTILITARIAN or the objective GROUP-EGALITARIAN with our set of constraints being (7). Looking at the form of the GROUP-UTILITARIAN and the GROUP-EGALITARIAN objectives, it is not difficult to see that they are linear (and therefore convex) in the parameters $x_{ij}$ and $\Delta_h$, however as the following theorem shows, the constraint set (7) is not convex. In fact, either of the proportion bounds alone in constraint (7d) would lead to a non-convex set. The non-convexity of the constraint set implies that the resulting optimization problem would also be non-convex:

**Theorem 6.3.** *The constraint set (7) is not convex.*

Although the constraint set (7) is not convex, if we fix the values of $\Delta_h$ then we clearly have a simple feasibility LP with variables $x_{ij}$. We therefore take an approach where for a given objective (GROUP-UTILITARIAN or GROUP-EGALITARIAN), we search for the corresponding optimal values of $\Delta_h$ by running the feasibility LP of (7). Note that with a given set of values for $\Delta_h$, we can obtain the corresponding value for the GROUP-UTILITARIAN or GROUP-EGALITARIAN objectives and therefore the LP does not need an objective: a feasibility check suffices. Further, since we only use non-trivial values for $\Delta_h \in [0,1]$, constraint (7c) can be omitted. Sections 6.1.1 and 6.1.2 detail how we use the feasibility LPs of (7) to obtain LP solutions that are approximately optimal (having bounded additive approximation from the optimal) for the GROUP-UTILITARIAN and GROUP-EGALITARIAN objectives, respectively. Since these resulting LP solutions could contain fractional values, i.e., it is possible to have a value $x_{ij} \notin \{0,1\}$, the approximately optimal LP solution would have to be rounded to an integral solution. This rounding further degrades the approximation, but we show that this degradation is not large and can be bounded. The details of the rounding are shown

in Section 6.1.3. The search algorithms of Sections 6.1.1 and 6.1.2, followed by the rounding of Section 6.1.3, lead to an algorithm for **FABC**.

### 6.1.1 Search Algorithm for the GROUP-UTILITARIAN Objective

We are searching for the optimal proportional violations $\Delta_h^* \in [0, 1]$ for the GROUP-UTILITARIAN objective. The first step we take is to discretize the space by a parameter $\epsilon \in (0, 1)$. For convenience, we set $\epsilon = \frac{1}{r}$ where $r \in \mathbb{Z}^+$, i.e., $r$ is a positive integer. Accordingly, instead of interacting with the continuous interval $[0, 1]$ for the proportional violations, we instead interact with $E_\epsilon = \{\epsilon, 2\epsilon, \ldots, \ldots, (\frac{1}{\epsilon} - 1)\epsilon, 1\}$, with $|E_\epsilon| = \frac{1}{\epsilon}$. Therefore, we have a set of $(\frac{1}{\epsilon})^{|\mathcal{H}|}$ many possibilities for the proportional violations and we can obtain the optimal solution for the GROUP-UTILITARIAN objective through exhaustive search by checking the feasibility of LP (7) and picking the feasible combination of proportional violations which leads to the smallest value for the GROUP-UTILITARIAN objective, i.e., GROUP-UTILITARIAN $= \sum_{h \in \mathcal{H}} \Delta_h$.

The above approach would take $O((\frac{1}{\epsilon})^{|\mathcal{H}|})$ many LP runs. Therefore, we show a faster search that tries instead $O((\frac{1}{\epsilon})^{|\mathcal{H}|-1})$. The key to this speed up comes from the fact that for the two-color case, we only need to evaluate $O(\frac{1}{\epsilon})$ many possibilities.

**Theorem 6.4.** *For **FABC** with* GROUP-UTILITARIAN *objective, we can use* $O\left(\left(\frac{1}{\epsilon}\right)^{|\mathcal{H}|-1}\right)$*–many LP runs to obtain an LP solution with additive approximation* $|\mathcal{H}|\epsilon$.

Furthermore, for the important two-color case with *symmetric* upper and lower bounds we show a search algorithm that requires only $O\left(\log \frac{1}{\epsilon}\right)$ LP runs. The two color case with *symmetric* upper and lower bounds is that where the two colors $h_1$ and $h_2$ are present with proportions $r_1$ and $r_2$ in the dataset, and the proportion bounds are set to $\alpha_i = r_i + \lambda_i, \beta_i = r_i - \lambda_i$ for $i \in \{1, 2\}$ and some valid $\lambda_1, \lambda_2 \in [0, 1]$. The key observation for the two-color symmetric case is that the proportion of one color implies the proportion of the other; hence, we can run binary search over the set $E_\epsilon$.

**Theorem 6.5.** *For **FABC** with two colors, symmetric lower & upper bounds, and the* GROUP-UTILITARIAN *objective, we can use* $O\left(\log(\frac{1}{\epsilon})\right)$*–many LP runs to get a solution with an additive approximation of* $|\mathcal{H}|\epsilon = 2\epsilon$.

### 6.1.2 Search Algorithm for GROUP-EGALITARIAN and GROUP-LEXIMIN Objectives

For the GROUP-EGALITARIAN objective we follow the same discretization step as for the GROUP-UTILITARIAN objective. For all colors, their violation $\Delta_h$ is set to the same value and the optimal solution is found simply by doing binary search over the set $E_\epsilon$ by running the feasibility LP (7).

**Theorem 6.6.** *For **FABC** with the* GROUP-EGALITARIAN *objective, we can use* $O\left(\log\left(\frac{1}{\epsilon}\right)\right)$*–many LP runs to get a solution with an additive approximation of* $\epsilon$.

We provide a heuristic algorithm for the GROUP-LEXIMIN objective; a rough sketch follows. In the first step, it obtains the GROUP-EGALITARIAN solution. Then, it proceeds by finding a color that cannot improve beyond the current optimal violation; if more than one color is found, then one of these colors is randomly picked. The algorithm then looks for the optimal violation for the remaining colors, having the violations of the previous colors fixed. These steps are followed until no color can have its proportional violation improved. See Appendix B for the full details.

### 6.1.3 The Rounding Scheme and ALG-FABC Guarantees

Having obtained the optimal LP solutions for either the GROUP-UTILITARIAN or GROUP-EGALITARIAN objectives, we now round the solutions to integral values at a bounded increase to the additive approximation. To do the rounding, we apply the network flow method of [11] (see Appendix E for details), although other rounding methods are applicable. Given the LP solution $x_{ij}$ and its associated proportional violations $\Delta_h$, if we denote the rounded integral solution by $\bar{x}_{ij}$ and $\bar{\Delta}_h$, then network-flow rounding guarantees the following: **(i)** $\sum_{i,j} d^p(i,j)\bar{x}_{ij} \le \sum_{i,j} d^p(i,j)x_{ij}$. **(ii)** $\forall i \in [k]: \left\lfloor \sum_{j \in \mathcal{C}} x_{ij} \right\rfloor \le \sum_{j \in \mathcal{C}} \bar{x}_{ij} \le \left\lceil \sum_{j \in \mathcal{C}} x_{ij} \right\rceil$. **(iii)** $\forall h \in \mathcal{H}, \forall i \in [k]: \left\lfloor \sum_{j \in \mathcal{C}^h} x_{ij} \right\rfloor \le \sum_{j \in \mathcal{C}^h} \bar{x}_{ij} \le \left\lceil \sum_{j \in \mathcal{C}^h} x_{ij} \right\rceil$.

Property **(i)** ensures that the clustering objective will not increase beyond the LP value, and thus, provided the LP cost does not exceed the upper bound on the cost $U$, the cost of the rounded assignment will not exceed $U$ as well. Property **(ii)** guarantees that the total number of points assigned to a cluster will not vary by more than 1 point. Property **(iii)** guarantees that the total number of points of a given color assigned to a cluster will not vary by more than 1 point. We can use the above properties along with with the lower bound on the size of any cluster $L(U)$ to establish the following theorem:

**Theorem 6.7.** *For the* **FABC** *problem, the rounded solution has cost of at most $U$ and an additive approximation of: (1) $|\mathcal{H}|(\epsilon + \frac{2}{L(U)})$ for the* GROUP-UTILITARIAN *objective and (2) $\epsilon + \frac{2}{L(U)}$ for the* GROUP-EGALITARIAN *objective.*

Recalling the additive approximation lower bounds of Theorem 5.3 for the **FABC** problem, we see that we obtain a solution for **FABC** of cost at most $U$ with near-optimal additive approximation. Specifically, our additive approximations for the GROUP-UTILITARIAN and GROUP-EGALITARIAN are $\frac{2|\mathcal{H}|}{L(U)}$ and $\frac{2}{L(U)}$ compared to their lower bounds of $\frac{|\mathcal{H}|}{8L(U)}$ and $\frac{1}{8L(U)}$, respectively.

**A randomized extension.** We also briefly mention a randomized rounding algorithm's guarantees; the description of this algorithm (which is motivated by an approach of [35]) is detailed in Appendix G. This algorithm efficiently constructs a random vector $\bar{X}$ with entries $\bar{X}_{i,j}$ in $\{0,1\}$ such that: (a) Properties (ii) and (iii) hold with probability 1, and (b) the expected value $E[\bar{X}_{i,j}]$ equals $x_{i,j}$ for all $(i,j)$. This has three consequences: (b1) the fairness guarantee for each cluster and color become *better in expectation*: for all $h \in \mathcal{H}$ and for all $i \in S$: $E[\sum_{j \in \mathcal{C}^h} \bar{X}_{ij}]$ indeed lies between $(\beta_h - \Delta_h)\left(\sum_{j \in \mathcal{C}} x_{ij}\right)$ and $(\alpha_h + \Delta_h)\left(\sum_{j \in \mathcal{C}} x_{ij}\right)$. (b2) The expected value of the objective function (the left-hand side in (i)) is at most the right-hand side of (i). (b3) Even if we had multiple linear objective functions, they will all be preserved in expectation.

# 7 Fairness Across the Clusters is not Possible

It is tempting to modify both the GROUP-UTILITARIAN and GROUP-EGALITARIAN (or GROUP-LEXIMIN) objectives to sum across the clusters instead of taking the maximum violation across the clusters. More specifically, we can replace the objectives by the following: GROUP-UTILITARIAN-SUM, which equals $\sum_{h \in \mathcal{H}, i \in [k]} \Delta_h^i$, and GROUP-EGALITARIAN-SUM, which equals $\min \max_{h \in \mathcal{H}} \sum_{i \in [k]} \Delta_h^i$, where $\Delta_h^i$ is the violation of color $h$ in cluster $i$; clearly the previously-considered violation $\Delta_h$ is $\max_{i \in [k]} \Delta_h^i$. It can be seen that such an objective is more flexible. For example, the maximum violations might occur in a cluster that cannot be improved within the given bound on the clustering cost, while it may be possible to improve it for other clusters. The original GROUP-UTILITARIAN and GROUP-EGALITARIAN objectives may bring no improvement in such a situation but their above modifications could. We prove a negative result. Specifically, while we were able to approximate **FABC** by small additive values for the original objectives (Theorem 6.7), for the new objectives we cannot efficiently approximate the **FABC** problems within even relatively-large additive approximations:

**Theorem 7.1.** *For* **FABC**, *the objectives* GROUP-UTILITARIAN-SUM *and* GROUP-EGALITARIAN-SUM *that sum across the clusters cannot be approximated in polynomial time to within an additive approximation of $O(n^\delta)$ where $\delta$ is a constant in $[0, 1)$, unless $P = NP$.*

# 8 Solving the Problem Optimally for a Large-Enough Cost

It seems reasonable to assume that when the upper bound on the cost $U$ is large enough, the problem becomes solvable in polynomial time. It is not difficult to devise such guarantees for some special cases. However, in the theorem below we show that an algorithm with such a guarantee would imply a true approximation[3] for fair clustering. Since fair clustering has remained resistant to a true polynomial-time approximation for general metric spaces and arbitrary lower- and upper- proportion bounds [13, 8], this suggests that the problem is indeed nontrivial. Furthermore, we also show the converse, i.e., a true approximation algorithm for fair clustering would imply an exact algorithm for fair clustering under a bounded cost **FCBC**.

---

[3]A true approximation algorithm yields a solution that approximates the optimal objective value with no constraint violation, in contrast to bi-criteria algorithms which have a bounded violation in the constraints.

**Theorem 8.1.** *Suppose that there is a polynomial time algorithm which can obtain the optimal solution for* **FCBC** *for the upper bound of $U$ if $U \geq \alpha(\mathcal{I}) \operatorname{OPT}_{cb}(\mathcal{I})$ where $\mathcal{I}$ is a specific instance of* **FCBC** *and $\operatorname{OPT}_{cb}(\mathcal{I})$ is the optimal cost of its color-blind clustering. Then we have a true polynomial time approximation algorithm for fair clustering. Further, a true polynomial time $\alpha'(\mathcal{I})$-approximation algorithm for fair clustering implies that* **FCBC** *can be solved optimally in polynomial time for $U \geq \alpha'(\mathcal{I}) \operatorname{OPT}_{\mathbf{FC}}(\mathcal{I})$.*

# 9 Experiments

We validate our algorithms on datasets from the UCI repository [24]. The results here are for $k$-means clustering; additional experiments are in Appendix F.

**Hardware, Software, and Algorithms:** We only use commodity hardware for all experiments with our programs running on Python 3.6. For the color-blind $k$-means clustering, we use the $k$-means++ algorithm [5] which has an approximation ratio of $O(\log k)$. Our LPs are solved using CPLEX [32]. Scikit-learn [46] is called for subroutines such as $k$-means++. The network-flow rounding is handled using NetworkX [25].

**Datasets:** We use all 32,561 entries of the **Adult** dataset [34]. For the **Census1990** dataset [41], because of its large size (over 2 million points) we sub-sample the dataset to a range similar to that considered in the fair clustering literature [20, 10]; specifically we use 20,000 data points. We also use the **CreditCard** dataset [47] which has 30,000 points (results are in Appendix F). For all datasets we use the numeric attributes to assign the coordinates in the space and the distance between any two points is set to the Euclidean distance.

**Setting and Measurements:** Each color $h \in \mathcal{H}$ has proportion $r_h$, i.e., $r_h = \frac{|\mathcal{C}^h|}{|\mathcal{C}|}$. We set the upper and lower bounds for each color to $\alpha_h = (1+\delta)r_h$ and $\beta_h = (1-\delta)r_h$. This means that each cluster should have each color with the same proportion as in the population with a possible deviation of $\delta$.

We first do color-blind clustering using the $k$-means++ algorithm. The clustering cost we obtain from the $k$-means++ is a proxy of the lowest possible value of the clustering cost (since the hardness of clustering forbids the calculation of the true optimal value). We gradually increase the upper bound cost from the color-blind cost to higher values and for each choice of the upper bound, we minimize either the GROUP-UTILITARIAN or GROUP-LEXIMIN objectives using our algorithms and record the objective value. For better interpretation, instead of showing the value of the upper bound, we show its ratio to the color-blind clustering, which is the PoF. Further, for all experiments we discretize the space by $\epsilon = \frac{1}{2^7} < 0.008$.

## 9.1 GROUP-UTILITARIAN Experiments

We use the **Adult** and **Census1990** datasets with self-reported gender (male or female) as the attribute. We note that both datasets explicitly use categorical labels for this socially-complex concept, and acknowledge that this is reductive [17]. Figure 2 shows the PoF versus the achieved GROUP-UTILITARIAN objective, with $\delta = 0.1$. As expected, as the price of fairness increases (higher bound on the cost), we can further minimize the proportional violations. Eventually the GROUP-UTILITARIAN objective becomes less than $0.1$ and even very close to zero. We also observe that at a given cost upper bound, we can achieve lower values for the GROUP-UTILITARIAN objective when the number of clusters ($k$) is lower.

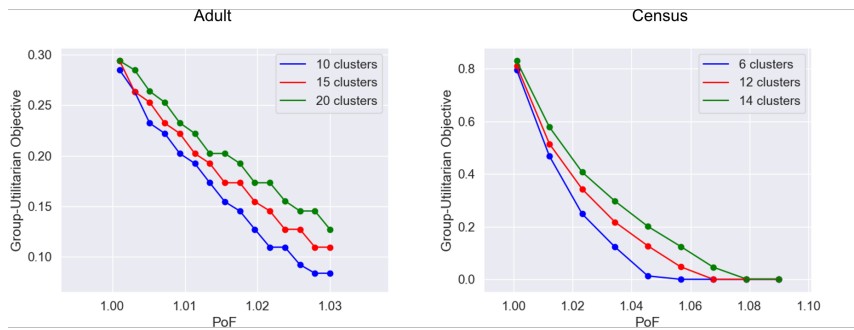

*Figure 2:* PoF *vs the* GROUP-UTILITARIAN *objective for the* ***Adult*** *and* ***Census1990*** *datasets.*

## 9.2  GROUP-EGALITARIAN and GROUP-LEXIMIN Experiments

We again use the **Adult** and **Census1990** datasets. However, for **Adult**, we set the fairness attribute to race which—in this dataset, and with the same inherent social caveats as the categorization of gender—has 5 groups (colors). For **Census1990**, we set the fairness attribute to age where we have three age groups.[4] We set $\delta = 0.05$ and $k = 10$ for **Adult** and $\delta = 0.1$ and $k = 5$ for **Census1990**. Figure 3 shows the results of our algorithm. We notice that for some colors smaller violations are harder to achieve and we need to set the maximum allowable clustering cost to larger values to reduce their violations.

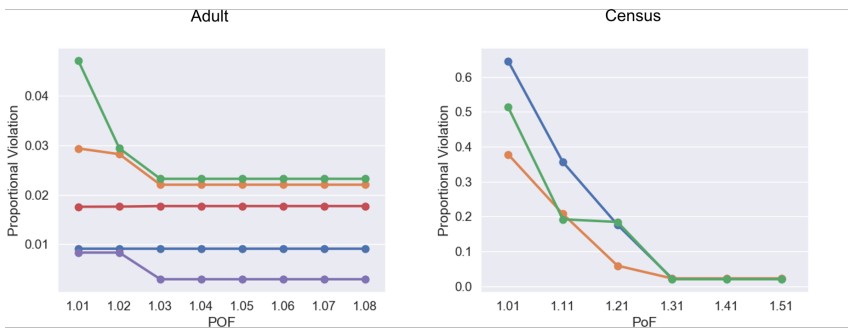

*Figure 3:* PoF *versus the proportional violation for different groups (each colored graph is a group) in the **Adult** and **Census1990** datasets.*

## 9.3  Checking the Size of the Smallest Cluster

As mentioned in Section 5 and Theorem 6.1 our approximations are dependent on the size of the smallest cluster in the solution. While it is not tractable to obtain the value of $L(U)$ for a given $U$, we can still empirically check the size of the smallest cluster in the cost bounded clusterings we obtain. We note that, throughout, we do not impose any lower bound on the cluster size in our algorithm. For the above experiments we considered, we find that the minimum cluster size (across all choices of $k$) are as follows: **Adult** (159 points), **Census1990** (171 points). The fact that the size of the smallest cluster is large means that we are achieving small (accurate) additive approximations with near-optimal objective values and when we obtain a large objective value it is because of how stringent the cost upper bound is.

**Acknowledgments.** Seyed Esmaeili and John Dickerson were supported in part by NSF CAREER Award IIS-1846237, NSF D-ISN Award #2039862, NSF Award CCF-1852352, NIH R01 Award NLM-013039-01, NIST MSE Award #20126334, DARPA GARD #HR00112020007, DoD WHS Award #HQ003420F0035, and a Google Faculty Research award. Aravind Srinivasan was supported in part by NSF awards CCF-1749864 and CCF-1918749, as well as research awards from Adobe, Amazon, and Google. We thank the anonymous reviewers for helpful feedback.

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
