# A Omitted Proofs

In this section, we provide proofs for theorems and lemmas in the main paper. We recall Theorem 5.1:

**Theorem 5.1.** *The fair clustering* **FC** *(1) and fair assignment* **FA** *(2) problems are NP-hard.*

*Proof.* Since fair clustering problems, i.e. fair $k$-(center, median, or means) generalize their NP-hard classical counterparts, i.e. the $k$-(center, median, or means) clustering, it follows that fair clustering problems are also NP-hard.

The hardness of the fair assignment problem was established by [11] for $k$-center clustering. Here we show that fair assignment is NP-hard for $k$-median and $k$-means clustering as well.

First, following Section 4 of [11], the reduction is from the Exact Cover by 3-Sets (X3C). In Exact Cover by 3-Sets, we have a universal set of elements $\mathcal{U}$ with $|\mathcal{U}| = 3r$ where $r$ is a positive integer and a set $\mathcal{F}$ whose elements are subsets of $\mathcal{U}$. The problem is to decide if there exists a set $\mathcal{F}'$ such that $\mathcal{F}' \subseteq \mathcal{F}$ and each element in $\mathcal{U}$ is included exactly once in one set in $\mathcal{F}'$.

The reduction is done by creating the following graph (see Figure 4 for an example). In the lowest level we have the elements $e$ of the set $\mathcal{U}$ each represented with a blue vertex. In the higher level we have the sets in $\mathcal{F}$ each represented with a blue vertex. We draw edges between vertices in $e \in \mathcal{U}$ and vertices in $F \in \mathcal{F}$ if and only if the element $e \in F$. For set $F$ in $\mathcal{F}$ we add 3 auxiliary blue vertices which are connected to it through an edge. Finally, we add a set $\mathcal{T}$ of red vertices where $|\mathcal{T}| = \frac{|\mathcal{U}|}{3} = r$ in the highest level where each of those vertices is connected through an edge to every vertex in the set $\mathcal{F}$.

The distance function puts a cost of zero if the distance is between identical vertices and a cost of one between vertices connected through an edge. For vertices with no edges between them, the distance is the minimum distance found according to this graph by calculating the minimum cost path. This means that the distance between the blue auxiliary vertices and a center which is not their parent center is 3 (the path from the vertex to the associated center to an element in $\mathcal{T}$, then the specified center).

In fair assignment, the set of centers is already chosen. We choose the set of centers to be the elements of $\mathcal{F}$. Therefore, the number of centers $k = r$. Further, it is clear that this is a two color problem, we set the lower and upper bounds for the red color to $\beta_{\text{red}} = \alpha_{\text{red}} = \frac{1}{4}$. It follows that $\beta_{\text{blue}} = \alpha_{\text{blue}} = \frac{3}{4}$, i.e. the ratio of red to blue vertices is $1 : 3$.

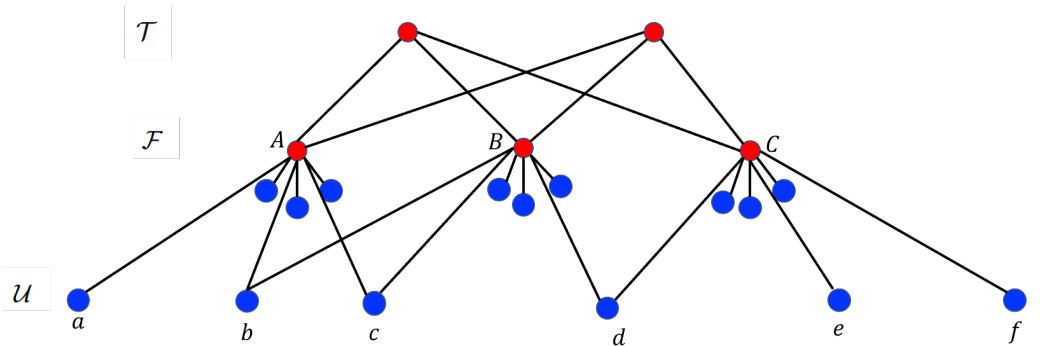

*Figure 4: Figure follows the example of [11]. We show the fair assignment resulting graph, from the given Exact Cover by 3-Sets example where we have $\mathcal{U} = \{a, b, c, d, e, f\}$ and $\mathcal{F} = \{A = \{a, b, c\}, B = \{b, c, d\}, C = \{d, e, f\}\}$.*

We note the following lemma:

**Lemma A.1.** *Given the constructed graph with the set of centers being $\mathcal{F}$, the minimum clustering cost is lower bounded by $1$ for the $k$-center problem and $n - k$ for the $k$-median and $k$-means.*

*Proof.* First we note the following fact:

**Fact A.1.** $\forall u, v \in G$ *where $u$ and $v$ are distinct, we have that $d(u, u) = d(v, v) = 0$ and $d(u, v) \geq 1$ if $u \neq v$.*

$k$**-center:** Since the number of points is greater than the number of centers it follows that there exists a point $u$ which will be assigned to another vertex $v$ and therefore $d(u, v) \geq 1$.

$k$**-median and $k$-means:** Denoting the assignment function (assigning vertices to centers) by $\phi$, the set of centers by $S$, and the integer $p$ where $p = 1$ for the $k$-median and $p = 2$ for the $k$-means, we have that:

$$\sum_{v \in G} d^p(v, \phi(v)) = \sum_{v \in S} d^p(v, \phi(v)) + \sum_{v \in G-S} d^p(v, \phi(v))$$

$$\geq 0 + \sum_{v \in G-S} d^p(v, \phi(v))$$

$$\geq 0 + \sum_{v \in G-S} 1^p$$

$$\geq \sum_{v \in G-S} 1$$

$$= n - k$$

where the above follows from Fact A.1. $\qquad\square$

Therefore, we have:

**Lemma A.2.** *If there exists an exact cover, then the fair assignment problem can have a $1 : 3$ red to blue vertex ratio and at a cost of 1 for the $k$-center and a cost of $n - k$ for the $k$-median and $k$-means.*

*Proof.* We translate the exact cover by 3-sets solution to the constructed graph. Each chosen set in exact cover $\mathcal{F}'$ will have the 3 corresponding elements from $\mathcal{U}$ assigned to its center, along with its 3 auxiliary vertices and 1 vertex from $\mathcal{T}$. If the set was not chosen in the exact cover, then it will have only its 3 auxiliary vertices assigned to it.

This clearly matches the lower bound on the cost function from lemma (A.1) for each clustering objective. Further, it is also clear that the $1 : 3$ red to blue color ratio is preserved in each cluster. $\quad\square$

**Lemma A.3.** *If there exists a fair assignment solution with $1 : 3$ red to blue proportion and whose cost is $1$ for the $k$-center and $(n - k)$ for the $k$-median and $k$-means, there exists a solution to the exact cover by 3-sets problem.*

*Proof.* The costs of 1 and $(n - k)$ for the $k$-center and $k$-median/mean respectively can only be achieved by assigning elements $e \in \mathcal{U}$ to a center that they have an edge between. Similarly, all of the blue auxiliary vertices have to be assigned to their parent. Further to achieve the $1 : 3$ red to blue ratio, a center will either choose 3 elements from $\mathcal{U}$ and therefore has to choose an element from $\mathcal{T}$ to satisfy the proportion. Or a center will not choose any element from $\mathcal{U}$ and in that case it would not need to pick an element from $\mathcal{T}$ to satisfy the proportion. $\quad\square$

$\square$

Here, we recall Theorem 5.2:

**Theorem 5.2.** *Fair clustering under a bounded cost* **FCBC** *and fair assignment under a bounded cost* **FABC** *are NP-hard.*

*Proof.* The hardness of fair clustering under a bounded cost **FCBC** simply follows by setting the upper bound to $U = \mathrm{OPT}_{\mathbf{FC}}$ where $\mathrm{OPT}_{\mathbf{FC}}$ is the optimal value of fair clustering **FC**. An optimal solution to fair clustering would achieve the optimal value of 0 for all possible fairness objectives of **FCBC** and would have a cost $\mathrm{OPT}_{\mathbf{FC}} \leq U$.

Conversely, an optimal solution for **FCBC** would have a proportional violation of zero for all colors (therefore it is fair). Moreover, its cost would not exceed $U = \text{OPT}_{\textbf{FC}}$. Therefore, it is an optimal solution for fair clustering.

By the above, a solution is optimal for a fair clustering if and only if it is an optimal solution to the corresponding **FCBC** instance with $U = \text{OPT}_{\textbf{FC}}$. It follows that since fair clustering is NP-hard from theorem (5.1), that fair clustering under a bounded cost **FCBC** is also NP-hard.

In a similar manner, by setting $U = \text{OPT}_{\textbf{FA}}$ the hardness of fair assignment under a bounded cost **FABC** can be established from the hardness of fair assignment. $\qquad\square$

Here, we recall Theorem 5.3:

**Theorem 5.3.** *For a given instance of the* **FCBC** *or* **FABC** *problem with an arbitrary upper bound* $U$, *unless* $P = NP$ *no polynomial time algorithm can produce a solution with a cost not exceeding* $U$ *that satisfies any of the following conditions: (a) The proportional violation of any color* $h \in \mathcal{H}$ *is* $\Delta_h < \frac{1}{8L(U)}$. *(b) The additive approximation for the* GROUP-UTILITARIAN *objective is less than* $\frac{|\mathcal{H}|}{8L(U)}$. *(c) The additive approximation for the* GROUP-EGALITARIAN *objective is less than* $\frac{1}{8L(U)}$.

*Proof.* We note that our derivation uses the reduction from X3C shown in the proof of theorem (5.1) and the resulting graph shown in figure (4). We start by deriving a collection of useful lemmas:

**Lemma A.4.** *If* $U = 1$ *for the* $k$-center *objective or* $U = n - k$ *for the* $k$-median *and* $k$-means *objectives, then* $L(U) = 4$ *for all objectives. Further the only set of centers that can lead to a cost not exceeding* $U$ *is* $S = \mathcal{F}$.

*Proof.* First it is clear that if we choose the set $\mathcal{F}$ to be the centers, i.e. $S = \mathcal{F}$, then if we route each point to one of its closest centers in $\mathcal{F}$, then we can have for the $k$-center we would have a cost of 1 since every point in the graph is at most a distance 1 from a point in $\mathcal{F}$. Further, for the $k$-median and $k$-means objectives, the points $\mathcal{F}$ would be routed to themselves and every other point would be routed to one of its closest centers in $\mathcal{F}$ which is at a distance of 1, this leads to a cost of $(0)k + (1)(n - k) = n - k$, therefore choosing the $\mathcal{F}$ as the set of centers we can indeed satisfy the upper bound $U$ for all objectives.

Now, consider another set of centers $S'$ such that $\exists i \in S'$ and $i \notin \mathcal{F}$, i.e. we have at least one center not from $\mathcal{F}$. Let $f$ be the point in $\mathcal{F}$ not selected in $S'$. For the $k$-center objective with $U = 1$, it follows that the blue auxiliary points of $f$ have to be made as centers since every other point is at least a distance of 2 away from them, but each auxiliary point of $f$ is made a center, then it follows that $|S' - \mathcal{F}| \geq 3$, i.e. at least two more points of $\mathcal{F}$ have not be selected as centers. We can invoke the argument again on the new auxiliary points to conclude that $|S' - \mathcal{F}| \geq 9$. Invoking the argument again, we will see get that $|S' - \mathcal{F}| \geq 3k$ which is infeasible since $|S' - \mathcal{F}| \leq |S'| \leq k$. Therefore, for the $k$-center with $U = 1$, we must have $S = \mathcal{F}$. Now having proven that $S = \mathcal{F}$ and since $U = 1$, it follows that the smallest cluster size is 4 formed by mapping the center in $S = \mathcal{F}$ to itself along with its auxiliary points, i.e. $L(U) = 4$ for the $k$-center.

For the $k$-median and $k$-means objectives with $U = n - k$, similiar to the $k$-center it is clear that every point which has not been selected as a center must have a center at a distance of at most 1 away. If we exclude one point $f \in \mathcal{F}$ from the set of centers, then its auxiliary points will each have to become centers to satisfy the upper bound cost of $U = n - k$, but this would mean that there are at least 2 more points in $\mathcal{F}$ that have been excluded. Following an argument similar to that of the $k$-center, we will have that the set of required centers would be at least $3k$ which is a contradiction. Therefore, the only possible choice of centers is $S = \mathcal{F}$. It follows as well that the smallest cluster size if 4 formed by mapping the center in $S = \mathcal{F}$ to itself along with its auxiliary points, i.e. $L(U) = 4$ for the $k$-median and $k$-means objectives. $\qquad\square$

Further, we define $\Delta^i_{\text{red}}$ and $\Delta^i_{\text{blue}}$ as the red and blue violations in the $i^{\text{th}}$ cluster, respectively. Then we have the following lemma

**Lemma A.5.** *For the two color case of the above reduction,* $\Delta^i_{red} = \Delta^i_{blue}$ *and* $\Delta_{red} = \Delta_{blue}$.

*Proof.* for cluster $i$, consider the red and blue violations $\Delta^i_{\text{red}}, \Delta^i_{\text{blue}}$ at that cluster, then we have:

$$\Delta^i_{\text{red}} = |p^i_{\text{red}} - \frac{1}{3}| = |(1 - p^i_{\text{blue}}) - (1 - \frac{2}{3})| = |\frac{2}{3} - p^i_{\text{blue}}| = \Delta^i_{\text{blue}}$$

It is clear then that $\Delta_{\text{red}} = \max_{i \in [k]} \Delta^i_{\text{red}} = \max_{i \in [k]} \Delta^i_{\text{blue}} = \Delta_{\text{blue}}$ $\qquad\square$

The following lemma follows immediately from the above:

**Lemma A.6.** *For the two color case of the above reduction* GROUP-UTILITARIAN $=$ 2GROUP-EGALITARIAN.

*Proof.* GROUP-UTILITARIAN $= \Delta_{\text{red}} + \Delta_{\text{blue}} = 2\Delta_{\text{red}} = 2$GROUP-EGALITARIAN. $\qquad\square$

We also note the following lemma:

**Lemma A.7.** *For a given cluster $i$ with set of points $C_i$, if the set of red points in the cluster $C^{red}_i$ satisfy $\Delta^i_{red} = |\frac{|C^{red}_i|}{|C_i|} - \frac{1}{4}| < \frac{1}{4|C_i|}$, then cluster $i$ has no violation.*

*Proof.* Suppose that $|\frac{|C^{\text{red}}_i|}{|C_i|} - \frac{1}{4}| < \frac{1}{4|C_i|}$, then it follows that $||C^{\text{red}}_i| - \frac{1}{4}|C_i|| < \frac{1}{4}$. Since $|C_i|$ is an integer it follows that $\frac{1}{4}|C_i|$ is of the form $m, m + \frac{1}{4}, m + \frac{1}{2}$, or $m + \frac{3}{4}$ where $m$ is an integer. Further since $|C^{\text{red}}_i|$ is also an integer, the fact that $||C^{\text{red}}_i| - \frac{1}{4}|C_i|| < \frac{1}{4}$ implies that $|C^{\text{red}}_i| = \frac{1}{4}|C_i|$ and we have no violation for the red color in cluster $i$. Further, from Lemma A.5 the blue violation equals the red violation and therefore we have no violation in cluster $i$. $\qquad\square$

Now we are ready to prove the main claims for the **FCBC** problem.

For the first claim, assume by contradiction that a polynomial time algorithm gave a solution of violation less than $\frac{1}{8L}$ and cost $\leq U$. Now, if we consider clusters $i$ of size $|C_i|$ such that $4 \leq |C_i| \leq 8$, then it clear that since $\Delta^i_{\text{red}} \leq \Delta_{\text{red}} \leq \frac{1}{8L(U)}, \Delta^i_{\text{red}} \leq \frac{1}{4|C_i|}$ because $|C_i| \leq 8 \leq 2L(U)$, therefore there is no violation in these clusters by Lemma A.7.

Now consider a cluster of size greater than 8, (note by Lemma A.4 that $S = \mathcal{F}$) because of the upper bound $U$ such clusters could only add points for the top row set $\mathcal{T}$ to the cluster which are all red, it clear that the more red points are added the greater the violation, if one additional red point is added, then for the best color proportions the cluster has a total of: 6 blues and 3 reds, which lead to a violation of $|\frac{1}{3} - \frac{1}{4}| = \frac{1}{12} > \frac{1}{8L(U)} = \frac{1}{32}$, therefore it is impossible for the algorithm to form such clusters as that would contradict the assumption that the algorithm obtains a violation $< \frac{1}{8L(U)}$ for each color. Therefore such clusters are not possible. This means that there is no violation in any cluster and that the problem has been solved optimally which by the NP-hardness is impossible unless $P = NP$.

Now the two remaining claims follow easily. By definition we have that GROUP-EGALITARIAN $=$ $\max_{h \in \mathcal{H}} \Delta_h$. If GROUP-EGALITARIAN $< \frac{1}{8L(U)}$, then it follows that $\Delta_h < \frac{1}{8L(U)}$ for every color $h \in \mathcal{H}$ which by the first claim cannot happen unless $P = NP$.

Further, by Lemma A.6 GROUP-UTILITARIAN $=$ 2 GROUP-EGALITARIAN, therefore if GROUP-UTILITARIAN $< \frac{|\mathcal{H}|}{8L(U)}$, then GROUP-EGALITARIAN $< \frac{1}{8L}$. which is impossible unless $P = NP$.

The same claims for the **FABC** problem can be proven by simply setting the set of centers $S = \mathcal{F}$ and the upper bound $U = 1$ for $k$-center and $n - k$ for the $k$-median/means, then following similar arguments. $\qquad\square$

**Note:** In the proof above, if we consider only the upper bound cost $U$ and ignore the fairness objective the problem is solvable in polynomial time. For **FCBC** simply set $S = \mathcal{F}$, then route each point to one of its closest centers. For the **FABC** with a given $S = \mathcal{F}$, again simply route each points to one of its closest centers. This highlights that the hardness is not from the upper bound cost $U$.

Next, we recall Theorem 6.2

**Theorem 6.2.** *For any clustering objective and both the* GROUP-UTILITARIAN *and* GROUP-EGALITARIAN *objectives, given an algorithm that solves fair assignment under a bounded cost* **FABC** *with additive approximation $\mu$, the fair clustering under a bounded cost* **FCBC** *problem can be solved with an additive approximation of $\mu$ and at a cost of at most $(2 + \alpha)U$, where $\alpha$ is the approximation ratio of the color-blind clustering algorithm.*

*Proof.* Let $S$ and $\phi$ be the set of centers and assignment of the color-blind algorithm. Let $S^*$ and $\phi^*$ be the optimal set of centers and assignment for the fair assignment under bounded cost **FABC**. Let $\phi'$ be an assignment that routes the vertices from their center in $S^*$ to the nearest center in $S$, i.e. for a given vertex $j$, $\phi'(j) = \text{argmin}_{i' \in S} d(i', \phi^*(j))$. Based on this setting we can upper bound the objective based on the following:

$$d(j, \phi'(j)) \leq d(j, \phi^*(j)) + d(\phi'(j), \phi^*(j))$$
$$\leq d(j, \phi^*(j)) + d(\phi(i), \phi^*(j))$$
$$\leq d(j, \phi^*(j)) + d(j, \phi^*(j)) + d(j, \phi(j))$$
$$\leq 2d(j, \phi^*(j)) + d(j, \phi(j))$$

It follows then by the triangle inequality of the $p$-norm and the non-negativity of the components, that $\left( \sum_{j \in \mathcal{C}} d^p(j, \phi'(j)) \right)^{1/p} \leq 2 \left( \sum_{j \in \mathcal{C}} d^p(j, \phi^*(j)) \right)^{1/p} + \left( \sum_{j \in \mathcal{C}} d^p(j, \phi(j)) \right)^{1/p} \leq 2U + \alpha U = (2 + \alpha)U$. Note that in the last inequality the bounded the color-blind cost as follows: $\left( \sum_{j \in \mathcal{C}} d^p(j, \phi(j)) \right)^{1/p} \leq \alpha \, \text{OPT}_{\text{cb}} \leq \alpha U$, where as noted the optimal color-blind cost $\text{OPT}_{\text{cb}}$ is upper bounded by $U$, i.e. $\text{OPT}_{\text{cb}} \leq U$ otherwise the problem would not be feasible. This proves the upper bound on the objective.

Now we establish guarantees on the proportions. For a given center $s$ in $S$, let $N(s) = \{i' \in S^* | s = \text{argmin}_{i \in S} d(i, i')\}$, i.e. $N(s)$ is the set of centers in $S^*$ routing their vertices to $s$. Denote the set of points assigned to cluster $i'$ by $\phi^{*-1}(i')$, i.e. $\phi^{*-1}(i') = \{j \in \mathcal{C} \, | \phi^*(j) = i'\}$. Then for any color $h$ we have that:

$$\min_{i' \in N(s)} \frac{\left( \sum_{j \in \phi^{*-1}(i'), \chi(j)=h} 1 \right)}{|\phi^{*-1}(i')|} \leq$$
$$\frac{\sum_{i' \in N(s)} \left( \sum_{j \in \phi^{*-1}(i'), \chi(j)=h} 1 \right)}{\sum_{i' \in N(s)} |\phi^{*-1}(i')|} \leq$$
$$\max_{i' \in N(s)} \frac{\left( \sum_{j \in \phi^{*-1}(i'), \chi(j)=h} 1 \right)}{|\phi^{*-1}(i')|}$$

That is the final color proportion will be within the lower and upper proportions of the routing centers. It follows that $\Delta_h$ does not increase for any color and that the GROUP-UTILITARIAN, GROUP-EGALITARIAN, and GROUP-LEXIMIN objectives using $\phi'$ are not greater than that of the optimal solution.

The above facts, combined with the premise of having an algorithm that solves the fair assignment under bounded cost **FABC** with an additive violation of $\mu$ completes the proof. $\square$

Next, we recall Theorem 6.3

**Theorem 6.3.** *The constraint set (7) is not convex.*

*Proof.* The non-convexity of the constraint set (7) can be shown even when ignoring the upper proportionality constraint, i.e. constraint (7d) only with the lower bound. Specifically, we would have

the following constraint set:

$$\sum_{i,j} d^p(i,j)x_{ij} \leq U^p \tag{8a}$$

$$\forall j \in \mathcal{C} : \sum_{i \in S} x_{ij} = 1, \quad x_{ij} \in [0,1] \tag{8b}$$

$$\forall h \in \mathcal{H} : \Delta_h \in [0,1] \tag{8c}$$

$$\forall h \in \mathcal{H}, \forall i \in S : (\beta_h - \Delta_h)\Big(\sum_{j \in \mathcal{C}} x_{ij}\Big) \leq \sum_{j \in \mathcal{C}^h} x_{ij} \tag{8d}$$

Now, assume that the upper bound on the cost $U$ is sufficiently large (this would let assignments of a high cost remain feasible). Consider the case of two colors: red and blue, with $\beta_{\text{red}} = \beta_{\text{blue}} = \frac{1}{2}$. Let each color constitute half the dataset, i.e. $|\mathcal{C}^{\text{red}}| = |\mathcal{C}^{\text{blue}}| = \frac{n}{2}$, clearly $|\mathcal{C}| = 2|\mathcal{C}^{\text{red}}| = 2|\mathcal{C}^{\text{blue}}| = n$. Set the number of clusters to two ($k = 2$), consider the following two feasible solutions $x_{ij}^1, \Delta_{\text{red}}^1, \Delta_{\text{blue}}^1$ and $x_{ij}^2, \Delta_{\text{red}}^2, \Delta_{\text{blue}}^2$ with $\Delta_{\text{blue}}^1 = \Delta_{\text{blue}}^2 = 1$, then the following holds (note that $\alpha = \frac{2}{3}$):

**For $x_{ij}^1, \Delta_{\text{red}}^1$:**

cluster 1: $\displaystyle\sum_{j \in \mathcal{C}^{\text{red}}} x_{1j}^1 = \sum_{j \in \mathcal{C}^{\text{blue}}} x_{1j}^1 = \alpha\frac{n}{2} = \frac{2}{3}\frac{n}{2} = \frac{n}{3}$

cluster 2: $\displaystyle\sum_{j \in \mathcal{C}^{\text{red}}} x_{2j}^1 = \sum_{j \in \mathcal{C}^{\text{blue}}} x_{2j}^1 = (1-\alpha)\frac{n}{2} = \frac{1}{3}\frac{n}{2} = \frac{n}{6}$

$\displaystyle |C_2| = \sum_{j \in \mathcal{C}} x_{2j}^1 = \frac{n}{3}$

$\Delta_{\text{red}}^1 = 0$

**For $x_{ij}^2, \Delta_{\text{red}}^2$:**

cluster 1: $\displaystyle\sum_{j \in \mathcal{C}^{\text{red}}} x_{1j}^2 = \frac{n}{2},$

$\displaystyle\sum_{j \in \mathcal{C}^{\text{blue}}} x_{1j}^2 = (1 - (\alpha + \frac{1}{n/2}))\frac{n}{2} = \frac{n}{6} - 1$

cluster 2: $\displaystyle\sum_{j \in \mathcal{C}^{\text{red}}} x_{2j}^2 = 0,$

$\displaystyle\sum_{j \in \mathcal{C}^{\text{blue}}} x_{2j}^2 = (\alpha + \frac{1}{n/2})\frac{n}{2} = (\frac{2}{3} + \frac{1}{n/2})\frac{n}{2} = \frac{n}{3} + 1$

$\displaystyle |C_2| = \sum_{j \in \mathcal{C}} x_{2j}^2 = \frac{n}{3} + 1$

$\Delta_{\text{red}}^2 = \dfrac{1}{2}$

We now form a simple convex combination of the two solutions $x_{ij} = \frac{1}{2}(x_{ij}^1 + x_{ij}^2), \Delta_{\text{red}} = \frac{1}{2}(\Delta_{\text{red}}^1 + \Delta_{\text{red}}^2) = \frac{1}{4}$. Constraints (8a), (8b), and (8c) would clearly be satisfied, but if we consider constraint (8d) for the red color and the second cluster, then we have:

$$RHS = \sum_{j \in \mathcal{C}^{\text{red}}} x_{2j} = \frac{n}{12}$$

$$LHS = (\frac{1}{2} - \frac{1}{4})(\frac{n}{3} + \frac{1}{2}) = \frac{n}{12} + \frac{1}{8}$$

It is clear that $LHS \leq RHS$ does not hold and therefore, the constraint is not satisfied for the convex combination and therefore the constraint set of the problem is indeed not convex.

A similar assignment of solutions can be used to show that the set is not convex if we were to consider only the over-representation constraint in (7d) instead. □

Next, we recall Theorem 6.4

**Theorem 6.4.** *For* **FABC** *with* GROUP-UTILITARIAN *objective, we can use* $O\left(\left(\frac{1}{\epsilon}\right)^{|\mathcal{H}|-1}\right)$ *–many LP runs to obtain an LP solution with additive approximation* $|\mathcal{H}|\epsilon$.

Before we give the proof of Theorem 6.4 we introduce some observations and algorithm (3). The core of the speed up is an algorithm that runs $O(\frac{1}{\epsilon})$ many LPs for the two color case. Therefore given a general instance of $|\mathcal{H}| > 2$ many colors, we simply explore all $O((\frac{1}{\epsilon})^{|\mathcal{H}|-2})$ possibilities for $|\mathcal{H}| - 2$ colors and find the optimal value for the two excluded colors through $O(\frac{1}{\epsilon})$ many LP runs for each possibility, leading to a total of $O((\frac{1}{\epsilon})^{|\mathcal{H}|-1})$.

**Algorithm for FABC with *two colors* and *arbitrary proportion bounds*:** Now we explain our algorithm for the two color case that runs at most $O(\frac{1}{\epsilon})$ many LPs. The algorithm utilizes two facts:

**Fact A.2.** *If the LP is feasible for* $\Delta_1$ *and* $\Delta_2$*, then it is also feasible for* $\Delta_1'$ *and* $\Delta_2'$ *where it either holds that* $\Delta_1' \geq \Delta_1$*,* $\Delta_2' \geq \Delta_2$*, or both.*

**Fact A.3.** *If the LP is feasible for* $\Delta_1$ *and* $\Delta_2$*, then* $\Delta_1' = \Delta_1 + i$ *and* $\Delta_2' = \Delta_2 - i$ *result in the same value for the* GROUP-UTILITARIAN *objective.*

As mentioned before we discretize the set of possibilities for $\Delta_1$ and $\Delta_2$ in the range $[0,1]$ by $\epsilon = \frac{1}{r}$ where $r$ is a positive integer. This leads to a set of $E_\epsilon \times E_\epsilon$ many possibilities where $E_\epsilon = \{\epsilon, 2\epsilon, \ldots, \ldots, (\frac{1}{\epsilon} - 1)\epsilon, 1\}$. This results in a two dimensional grid with each cell having a certain value for the GROUP-UTILITARIAN objective as figure 5 shows for a specific value of $\epsilon$.

The algorithm is shown in block (3). Note that $LP(\Delta_1, \Delta_2)$ is a function that returns **true** if the LP is feasible for the proportional violations of $\Delta_1$ and $\Delta_2$ for colors 1 and 2, respectively and returns **false** otherwise. The rough sketch of the algorithm is that it starts at the top right of the grid and checks for feasibility. It then chooses smaller values of $\Delta_1$ until it encounters an infeasible instance. Once that happens the algorithm moves diagonally, looking for a feasible instance[5]. Once a feasible instance is found on the diagonal, the algorithm moves vertically down until it reaches the bottom and terminates or reaches an infeasible instance and therefore goes back to exploring diagonally again. See figure 6 for a run of the algorithm.

**Lemma A.8.** *Algorithm (3) runs at most* $O(\frac{1}{\epsilon})$ *many LPs and finds a solution with additive approximation* $2\epsilon$.

*Proof.* First we start by bounding the total number of LP runs. Since the algorithm in each step either moves horizontally (at most $O(\frac{1}{\epsilon})$ steps), or diagonally (at most $O(\frac{1}{\epsilon})$ steps), or vertically (at most $O(\frac{1}{\epsilon})$ steps), the total number of LP runs is indeed $O(\frac{1}{\epsilon})$. We note that the horizontal and vertical explorations can be done faster through binary search but the diagonal still has to be done through a linear sweep which makes the asymptotic number of LP runs still $O(\frac{1}{\epsilon})$.

Now we prove that we obtain a values at most $2\epsilon$ greater than the optimal. Suppose that the optimal proportional violations are $\Delta_1^*$ and $\Delta_2^*$, then in the discrete grid we have $\bar{\Delta}_1$ and $\bar{\Delta}_2$ such that $\bar{\Delta}_1 - \Delta_1^* \leq \epsilon$ and $\bar{\Delta}_2 - \Delta_2^* \leq \epsilon$ which must be feasible, this follows by Fact (A.2). It is therefore clear that there is a cell in the gird with value at most $2\epsilon$ more than the optimal. Now we show that the algorithm either finds it or finds a better solution. If $\bar{\Delta}_2 = 1$, then during the horizontal exploration the algorithm will pass over the $(\bar{\Delta}_1, \bar{\Delta}_2)$ cell and therefore it will either output $\bar{\Delta}_1 + \bar{\Delta}_2$ as the optimal value or find a smaller value.

If $\bar{\Delta}_2 < 1$, then by Fact (A.2) cell $(\bar{\Delta}_1, 1)$ must be feasible. Since $(\bar{\Delta}_1, 1)$ is feasible, then if $(\bar{\Delta}_1 - \epsilon, 1)$ is infeasible the algorithm will switch to vertical exploration and encounter cell $(\bar{\Delta}_1, \bar{\Delta}_2)$. If a cell $(\Delta_1, 1)$ with $\Delta_1 < \bar{\Delta}_1$ is feasible, then the algorithm will continue horizontally exploring, assuming cell $(\epsilon, \epsilon)$ is not feasible[6] then the algorithm eventually switches to diagonal exploration

---

[5]Note that there is no point to explore above the diagonal as these are cells that lead to higher not lower values for the GROUP-UTILITARIAN objective

[6]If cell $(\epsilon, \epsilon)$ is feasible, then the algorithm will explore horizontally to the last cell, followed by vertical exploration to the last cell.

**Algorithm 3** Algorithm for Arbitrary Proportion Two Color Case for the GROUP-UTILITARIAN Objective

1: Set UTIL$^* = 2$, $\Delta_1^* = 1$, $\Delta_2^* = 1$.
2: Set direction $= 0$. {0 for horizontal, 1 for diagonal, 2 for vertical}
3: **while** (direction $= 1$ **and** $\Delta_1 \leq 1$ **and** $\Delta_2 \geq \epsilon$) **or** (direction $= 2$ **and** $\Delta_2 \geq \epsilon$) **do**
4:     **if** direction $= 0$ **then**
5:         $\Delta_1 = \Delta_1 - \epsilon$
6:         **if** $\Delta_1 = 0$ **then**
7:             $\Delta_1 = \Delta_1 + \epsilon$
8:             direction $= 2$
9:         **else if** $LP(\Delta_1, \Delta_2) =$ **TRUE then**
10:             $\Delta_1^* = \Delta_1$
11:             UTIL$^* = \Delta_1 + \Delta_2$
12:         **else if** $LP(\Delta_1, \Delta_2) =$ **FALSE then**
13:             $\Delta_1 = \Delta_1 + \epsilon$
14:             direction $= 1$ {Explore diagonally}
15:         **end if**
16:     **else if** direction $= 1$ **then**
17:         $\Delta_1 = \Delta_1 + \epsilon$
18:         $\Delta_2 = \Delta_2 - \epsilon$
19:         **if** $LP(\Delta_1, \Delta_2) =$ **TRUE then**
20:             UTIL$^* = \Delta_1 + \Delta_2$
21:             $\Delta_1^* = \Delta_1$
22:             $\Delta_2^* = \Delta_2$
23:             direction $= 2$ {Explore vertically}
24:         **end if**
25:     **else if** direction $= 2$ **then**
26:         $\Delta_2 = \Delta_2 - \epsilon$
27:         **if** $LP(\Delta_1, \Delta_2) =$ **TRUE then**
28:             UTIL$^* = \Delta_1 + \Delta_2$
29:             $\Delta_2^* = \Delta_2$
30:         **else**
31:             direction $= 1$ {Explore diagonally}
32:         **end if**
33:     **end if**
34: **end while**
35: **return** $\Delta_1^*$, $\Delta_2^*$, UTIL$^*$.

if this diagonal has cells of GROUP-UTILITARIAN value less than $\bar{\Delta}_1 + \bar{\Delta}_2$, then the algorithm has an encountered a feasible cell with value less than or equal to $\bar{\Delta}_1 + \bar{\Delta}_2$. If the diagonal has cells of GROUP-UTILITARIAN value equal to $\bar{\Delta}_1 + \bar{\Delta}_2$, then either the algorithm will encounter the cell $(\bar{\Delta}_1, \bar{\Delta}_2)$ or it will encounter another feasible with the same value by Fact A.3. If the diagonal has cells of GROUP-UTILITARIAN value greater than $\bar{\Delta}_1 + \bar{\Delta}_2$, then we note that we could have a collection of optimal cells of GROUP-UTILITARIAN value less than or equal to $\bar{\Delta}_1 + \bar{\Delta}_2$ located below the diagonal, then it should be clear that the algorithm will encounter the optimal cell with smallest $\Delta_1$ value. $\qquad\square$

Now we proof theorem Theorem 6.4:

*Proof of Theorem 6.4.* We have $|\mathcal{H}|$ many proportional violation values to decide, one for each color. We do exhaustive search over the proportional violation values in the discrete set $E_\epsilon$ for all colors except for the first two colors $h_1$ and $h_2$. For a given value of proportional violations for the colors in $\mathcal{H} - \{h_1, h_2\}$, we find the optimal values for colors $h_1$ and $h_2$ by running algorithm (3). It follows by Theorem (A.8), that we would find the optimal value for $h_1$ and $h_2$. Further, since finding the optimal value for the two colors $h_1$ and $h_2$ takes $O(\frac{1}{\epsilon})$ many LP runs and exhaustive search over the set of colors $\mathcal{H} - \{h_1, h_2\}$ takes $O((\frac{1}{\epsilon})^{|\mathcal{H}|-2})$ LP runs, then the optimal value can be found in at most $O((\frac{1}{\epsilon})^{|\mathcal{H}|-1})$ LP runs. $\qquad\square$

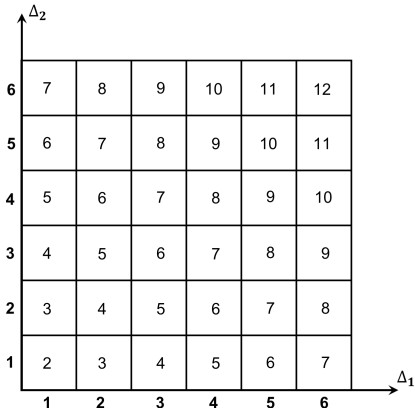

*Figure 5: The two dimensional grid formed by discretizing the values of $\Delta_1$ and $\Delta_2$. Here $\epsilon = \frac{1}{6}$. Note that each number shown is a multiple of $\epsilon$, i.e. the first coordinate on the ($\Delta_1$) axis is $1 \times \epsilon$, then $2 \times \epsilon$ and so on. For each cell we put the GROUP-UTILITARIAN objective value, which is again a multiple of $\epsilon$, i.e. the top right cell has a GROUP-UTILITARIAN of $12\epsilon = 12\frac{1}{6} = 2$. Notice how the cells on the same diagonal have the same value as stated in Fact (A.3.)*

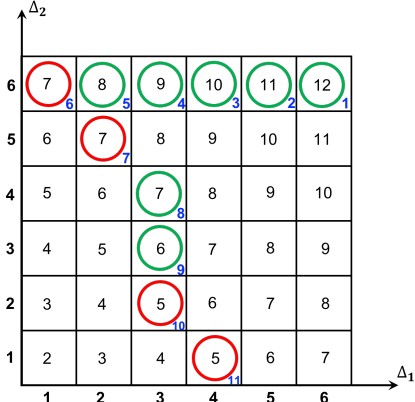

*Figure 6: Here we show a possible run of the algorithm over the grid of figure 5. The circled cells are ones where the LP was run. A green circle indicates that the LP was feasible whereas a red circle indicates that the LP was infeasible. The blue number indicates the order where the LP corresponding to the cell was run, accordingly we start at the top right cell. It should be clear that the algorithm starts by horizontal exploration to the left, that it then moves to diagonal exploration which changes to vertical exploration once a feasible cell is found. Furthermore, the vertical exploration changes to diagonal once a an infeasible cell is encountered. It should be clear that the optimal value found by the algorithm is at the cell with $\Delta_1 = \Delta_2 = 3\epsilon$ with GROUP-UTILITARIAN $= 6\epsilon$.*

Next, we recall Theorem 6.5

**Theorem 6.5.** *For* **FABC** *with two colors, symmetric lower & upper bounds, and the* GROUP-UTILITARIAN *objective, we can use* $O\left(\log(\frac{1}{\epsilon})\right)$*–many LP runs to get a solution with an additive approximation of* $|\mathcal{H}|\epsilon = 2\epsilon$.

Before we prove Theorem 6.5 we point out the following definition and observations. For the two color case, our color set is $\mathcal{H} = \{h_1, h_2\}$. Further, we denote the proportions for color $i$ by $r_i$ where $r_i = \frac{|\{j | j \in \mathcal{C}, \chi(j) = i\}|}{|\mathcal{C}|} = \frac{|\mathcal{C}^i|}{|\mathcal{C}|}$. We use color $h_1$ to denote the color with less points, i.e. $r_1 \leq r_2$. The upper and lower bounds we consider for each color are: $\beta_i = (1 - \delta)r_i$ and $\alpha_i = (1 + \delta)r_i$. The idea behind the algorithm is that the proportions of one color imply the proportion of the other color.

**Algorithm for FABC with *two colors* and *symmetric lower and upper proportion bounds*:** Our algorithm is based on the simple observation shown in figure 7

*Figure 7: Proportions and bounds for two colors. $r_1 = 0.25, r_2 = 0.75, \lambda_i = \delta r_i$ for $i \in \{1, 2\}$ where $\delta = 0.1$. Notice how if color 1 violates the upper bound by having $p_1 = 0.3$, then we must have $p_2 = 0.7$, but color 2 is not violating. On the other hand, a violation for color 1 with $p'_1 = 0.4$ implies $p'_2 = 0.6$ which causes a violation for color 2.*

Without loss of generality, let $\lambda_1 \leq \lambda_2$, based on the observation in figure 7, we have the following lemma:

**Lemma A.9.** *If $\Delta_1 < \lambda_2 - \lambda_1$, then $\Delta_2 = 0$. If $\Delta_1 \geq \lambda_2 - \lambda_1$, then $\Delta_2 = \Delta_1 - (\lambda_2 - \lambda_1)$*

*Proof.* Let color 1 have a $\Delta_1$ violating proportion of $p_1$, then in some cluster $p_1 = \alpha_1 + \Delta_1$ or $p_1 = \beta_1 - \Delta_1$.

Consider the case where $p_1 = \alpha_1 + \Delta_1$, then $p_2 = 1 - p_1 = 1 - \alpha_1 - \Delta_1$. Now if $\Delta_1 < \lambda_2 - \lambda_1$, then we have $p_2 > 1 - \alpha_1 - (\lambda_2 - \lambda_1) = 1 - \lambda_2 + \lambda_1 - \alpha = (1 - r_1) - \lambda_2 = r_2 - \lambda_2 = \beta_2$ this means that color 2 does not violate the lower bound. If we assume that color 2 violates the upper bound by an amount $\Delta_2 > 0$, then this would imply that $p_1 = 1 - p_2$ and the lower violation for color 1 would be $\beta_1 - p_1 = \beta_1 - (1 - \alpha_2 - \Delta_2) = \beta_1 - 1 + \alpha_2 + \Delta_2 = r_1 - \lambda_1 + r_2 + \lambda_2 + \Delta_2 - 1 = 1 + (\lambda_2 - \lambda_1) + \Delta_2 - 1 = (\lambda_2 - \lambda_1) + \Delta_2 > (\lambda_2 - \lambda_1)$ which is a contradiction since we assumed that $\Delta_1 < (\lambda_2 - \lambda_1)$.

Similarly, if $\Delta_1 \geq (\lambda_2 - \lambda_1)$, then we have $\Delta_2 = \beta_2 - (1 - \alpha_1 - \Delta_1) = \beta_2 - 1 + \alpha_1 + \Delta_1 = r_2 - \lambda_2 + r_1 + \lambda_1 - 1 + \Delta_1 = \lambda_1 - \lambda_2 + \Delta_1 = \Delta_1 - (\lambda_2 - \lambda_1)$. Now if we assume that color 2 has a violation of the upper bound by an amount $\Delta'_2 > \Delta_1 - (\lambda_2 - \lambda_1)$, this would imply that color 1 violates the lower bound by $\beta_1 - p_1 = r_1 - \lambda_1 + r_2 + \lambda_2 - 1 + \Delta_2 = (\lambda_2 - \lambda_1) + \Delta_2$ which is a contradiction since $\Delta_1 < \Delta_2 + (\lambda_2 - \lambda_1)$, therefore color 2 cannot violate by more than $\Delta_1 - (\lambda_2 - \lambda_1)$.

The case of $p_1 = \beta_1 - \Delta_1$ follows similar arguments. $\square$

The above observations lead to the following algorithm:

---

**Algorithm 4** GROUP-UTILITARIAN Algorithm for Two Colors with Symmetric Bounds for the GROUP-UTILITARIAN Objective

---

**Input:** set of points $\mathcal{C}$, price of fairness $U$, for each color $h \in \mathcal{H}$ lower and upper proportion values $\beta_h, \alpha_h$, error parameter $\epsilon$.
Define the set $E_\epsilon = \{0, \epsilon, \ldots, (\frac{1}{\epsilon} - 1)\epsilon\}$
Binary search $\Delta_1$ over the set $E_\epsilon$ by running the LP (7) ( if $\Delta_1 < \delta(r_2 - r_1)$ then $\Delta_2 = 0$, otherwise set $\Delta_2 = \Delta_1 - \delta(r_2 - r_1)$ ).

---

Now we are ready to prove Theorem 6.5.

*Proof of Theorem 6.5.* It follows from Lemma (A.9) that we can do binary search over the set $E_\epsilon$ using $\Delta_1$ as done in algorithm (4). Clearly, at most $O\left(\log(\frac{1}{\epsilon})\right)$ many LPs will be run because of binary search. Further, we know that we will find a solution at most $\epsilon$ greater, i.e. we worst case best LP value is: $\Delta_1^* + \epsilon, \Delta_2^* + \epsilon = (\Delta_1^* + \Delta_2^*) + 2\epsilon = \text{OPT} + 2\epsilon$. $\square$

Next, we recall Theorem 6.6

**Theorem 6.6.** *For* **FABC** *with the* GROUP-EGALITARIAN *objective, we can use* $O\left(\log\left(\frac{1}{\epsilon}\right)\right)-$ *many LP runs to get a solution with an additive approximation of $\epsilon$.*

*Proof.* For each color $h$ set $\Delta_h$ to the same value $\Delta$, do binary search over $E_\epsilon$ using LP(8). Clearly, the final value is at most $\epsilon$ greater than the optimal and we need at most $O(\log(\frac{1}{\epsilon}))$ many LP runs. $\square$

Now we introduce the following lemma:

**Lemma A.10.** $\bar{\Delta}_h < \Delta_h + \frac{2}{L(U)}$, *i.e. rounding will increase the violation by at most* $\frac{2}{L(U)}$.

*Proof.* Based on properties (ii) and (iii) from network flow rounding (see Section 6.1.3), we can get the following bound for the upper proportion:

$$
\begin{aligned}
\sum_{j\in\mathcal{C}^h} \bar{x}_{ij} &\leq \left\lceil \sum_{j\in\mathcal{C}^h} x_{ij} \right\rceil && \text{(by property (iii))}\\
&\leq \left\lceil \min\big((\alpha_h+\Delta_h),1\big)\Big(\sum_{j\in\mathcal{C}} x_{ij}\Big) \right\rceil && \text{(problem constraint)}\\
&< \min\big((\alpha_h+\Delta_h),1\big)\Big(\sum_{j\in\mathcal{C}} x_{ij}\Big) + 1 && \text{(ceiling upper bound)}\\
&\leq \min\big((\alpha_h+\Delta_h),1\big)\Big(\sum_{j\in\mathcal{C}} \bar{x}_{ij}+1\Big) + 1 && \text{(by property (ii))}\\
&\leq \min\big((\alpha_h+\Delta_h),1\big)\Big(\sum_{j\in\mathcal{C}} \bar{x}_{ij}\Big) + \min\big((\alpha_h+\Delta_h),1\big) + 1 && \\
&\leq \min\big((\alpha_h+\Delta_h),1\big)\Big(\sum_{j\in\mathcal{C}} \bar{x}_{ij}\Big) + 2 && \text{(since } \min\big((\alpha_h+\Delta_h),1\big)\leq 1\text{)}\\
&\leq (\alpha_h+\Delta_h)\Big(\sum_{j\in\mathcal{C}} \bar{x}_{ij}\Big) + 2 &&
\end{aligned}
$$

This implies that the new violation for the rounded solution $\bar{\Delta}_h$ satisfies:

$$
\alpha_h + \bar{\Delta}_h = \frac{\sum_{j\in\mathcal{C}^h} \bar{x}_{ij}}{\sum_{j\in\mathcal{C}} \bar{x}_{ij}} < \alpha_h + \Delta_h + \frac{2}{\sum_{j\in\mathcal{C}} \bar{x}_{ij}} \leq \alpha_h + \Delta_h + \frac{2}{L(U)}
$$

Therefore, we have:

$$
\bar{\Delta}_h - \Delta_h < \frac{2}{L(U)}
$$

For the lower proportions, we also have:

$$\sum_{j \in \mathcal{C}^h} \bar{x}_{ij} \geq \left\lfloor \sum_{j \in \mathcal{C}^h} x_{ij} \right\rfloor \qquad \text{(by property (iii))}$$

$$\geq \left\lfloor \max\big((\beta_h - \Delta_h), 0\big)\Big(\sum_{j \in \mathcal{C}} x_{ij}\Big) \right\rfloor \qquad \text{(problem constraint)}$$

$$> \max\big((\beta_h - \Delta_h), 0\big)\Big(\sum_{j \in \mathcal{C}} x_{ij}\Big) - 1 \qquad \text{(ceiling upper bound)}$$

$$\geq \max\big((\beta_h - \Delta_h), 0\big)\Big(\sum_{j \in \mathcal{C}} \bar{x}_{ij} - 1\Big) - 1 \qquad \text{(by property (ii))}$$

$$\geq \max\big((\beta_h - \Delta_h), 0\big)\Big(\sum_{j \in \mathcal{C}} \bar{x}_{ij}\Big) - \max\big((\beta_h - \Delta_h), 0\big) - 1$$

$$\geq \max\big((\beta_h - \Delta_h), 0\big)\Big(\sum_{j \in \mathcal{C}} \bar{x}_{ij}\Big) - 2 \qquad \text{(since } \max\big((\beta_h - \Delta_h), 0\big) \leq 1)$$

$$\geq (\beta_h - \Delta_h)\Big(\sum_{j \in \mathcal{C}} \bar{x}_{ij}\Big) - 2$$

$$\beta_h - \bar{\Delta}_h = \frac{\sum_{j \in \mathcal{C}^h} \bar{x}_{ij}}{\sum_{j \in \mathcal{C}} \bar{x}_{ij}} > \beta_h - \Delta_h - \frac{2}{\sum_{j \in \mathcal{C}} \bar{x}_{ij}} \geq \beta_h - \Delta_h - \frac{2}{L(U)}$$

Therefore, we have:

$$\bar{\Delta}_h - \Delta_h < \frac{2}{L(U)}$$

$\square$

Next, we recall Theorem 6.7:

**Theorem 6.7.** *For the* **FABC** *problem, the rounded solution has cost of at most $U$ and an additive approximation of: (1) $|\mathcal{H}|(\epsilon + \frac{2}{L(U)})$ for the* GROUP-UTILITARIAN *objective and (2) $\epsilon + \frac{2}{L(U)}$ for the* GROUP-EGALITARIAN *objective.*

*Proof.* (1) For the GROUP-UTILITARIAN, by theorems (6.4) and (6.5) the LP solution has a violation of $|\mathcal{H}| + \epsilon$, then by lemma (A.10) and the definition of the GROUP-UTILITARIAN $= \sum_{h \in \mathcal{H}} \Delta_h$, the violation is at most $|\mathcal{H}|(\epsilon + \frac{2}{L(U)})$.

(2) For the GROUP-EGALITARIAN, by theorem (6.6) and lemma (A.10), the rounded solution would have a worst case violation of $\epsilon + \frac{2}{L(U)}$ across the colors. $\square$

We introduce the following lemma which is important for proving Theorem 7.1:

**Lemma A.11.** *Any polynomial time approximation algorithm for* **FABC** *for a general upper bound $U$ must have $\mu > 0$, i.e. it must have a strictly greater than zero additive approximation guarantee.*

*Proof.* The proof follows from the proof of Theorem (5.2). Specifically, the proof of Theorem (5.2) shows that hard instances for **FABC** could have an optimal value of 0 for the GROUP-UTILITARIAN, GROUP-EGALITARIAN, and GROUP-LEXIMIN objectives, specifically when $U = \text{OPT}_{\text{FC}}$ where $\text{OPT}_{\text{FC}}$ is the optimal value of fair clustering. Therefore, if a polynomial time approximation algorithm with approximation ratio $\rho \geq 1$ and additive approximation $\mu \geq 0$ is ran over such hard instances, then it would output a solution of value $\rho \text{OPT} + \mu = \rho(0) + \mu = \mu$. If the algorithm has $\mu = 0$, then it would mean that the problem has been solved optimally which is impossible unless $P = NP$. Therefore, $\mu > 0$. $\square$

Now we recall Theorem 7.1:

**Theorem 7.1.** *For* **FABC***, the objectives* GROUP-UTILITARIAN-SUM *and* GROUP-EGALITARIAN-SUM *that sum across the clusters cannot be approximated in polynomial time to within an additive approximation of $O(n^\delta)$ where $\delta$ is a constant in $[0,1)$, unless $P = NP$.*

*Proof.* By the result of Lemma A.11 we know that we can hard instances with $\mathrm{OPT} = 0$ and that any polynomial time algorithm should have an additive approximation $\mu > 0$. Further, we consider the same X3C reduction of Theorem 5.2 and Figure 4 for **FABC** with the centers set to the points of $\mathcal{F}$.

To prove Theorem 7.1, suppose by contradiction that an algorithm $\mathcal{A}$ exists that guarantees an additive approximation of $O(n^\delta)$ for $\delta \in [0,1)$. Suppose, we are given an instance of the problem with optimal solution value of $\mathrm{OPT}$ and $n$ many points. Note by Lemma A.7 if $\Delta_{\text{red}}^i < \frac{1}{|C_i|}$, then there us no violation. It follows that if $\sum_{i \in [k]}(\Delta_{\text{red}}^i + \Delta_{\text{blue}}^i) < \frac{1}{4n}$ then we have no violation.

Now, create $D$ many duplicates of the given set of points. Let the distance between the points belonging to the same duplicate be the same as in the original instance, whereas for points in different duplicates the distance is infinity. Further, let the number of centers be $Dk$ where each duplicate has $k$ many centers assigned at the same points as the original instance. Given the original upper bound on the clustering objective $U$, the new upper bound $U'$ is set to $U' = U$ for the $k$-center, $U' = DU$ for the $k$-median, and $U' = \sqrt{D}U$ for the $k$-means objectives.

If this modified instance is given to $\mathcal{A}$, then the output would have a value of at most $\rho D \,\mathrm{OPT} + c(Dn)^\delta$ for some $c > 0$. If $D > \frac{1}{4^{\delta-1}} c^{\frac{1}{1-\delta}} n^{\frac{1+\delta}{1-\delta}}$ (which is polynomial in $n$), then the average violation across the duplicates is:

$$\frac{\rho D \,\mathrm{OPT} + c(Dn)^\delta}{D} = \rho \,\mathrm{OPT} + cn^\delta D^{\delta-1}$$

$$< \rho \,\mathrm{OPT} + c\frac{1}{4}n^\delta c^{\frac{\delta-1}{1-\delta}} n^{\frac{(1+\delta)(\delta-1)}{1-\delta}} = \rho \,\mathrm{OPT} + \frac{1}{4n} = 0 + \frac{1}{4n}$$

This means that there must exist at least one duplicate for which the violation is at most $\frac{1}{4n}$ which means that the problem has been exactly in polynomial time which is impossible unless $P = NP$. $\quad\square$

Next, we recall Theorem 8.1:

**Theorem 8.1.** *Suppose that there is a polynomial time algorithm which can obtain the optimal solution for* **FCBC** *for the upper bound of $U$ if $U \geq \alpha(\mathcal{I}) \,\mathrm{OPT}_{cb}(\mathcal{I})$ where $\mathcal{I}$ is a specific instance of* **FCBC** *and $\mathrm{OPT}_{cb}(\mathcal{I})$ is the optimal cost of its color-blind clustering. Then we have a true polynomial time approximation algorithm for fair clustering. Further, a true polynomial time $\alpha'(\mathcal{I})$-approximation algorithm for fair clustering implies that* **FCBC** *can be solved optimally in polynomial time for $U \geq \alpha'(\mathcal{I}) \,\mathrm{OPT}_{FC}(\mathcal{I})$.*

*Proof.* Suppose $\alpha(\mathcal{I}) \,\mathrm{OPT}_{cb}(\mathcal{I}) \leq \mathrm{OPT}_{FC}(\mathcal{I})$ where $\mathrm{OPT}_{FC}$ is the optimal fair clustering cost, then fair clustering is solvable in polynomial time which is impossible unless $P = NP$ since fair clustering is NP-hard.

If $\alpha(\mathcal{I}) \,\mathrm{OPT}_{cb}(\mathcal{I}) > \mathrm{OPT}_{FC}(\mathcal{I})$. Then this algorithm will not have any fairness violation if we choose $U = \alpha(\mathcal{I}) \,\mathrm{OPT}_{cb}(\mathcal{I})$, further its cost is $\alpha(\mathcal{I}) \,\mathrm{OPT}_{cb}(\mathcal{I}) \leq \alpha(\mathcal{I}) \,\mathrm{OPT}_{FC}(\mathcal{I})$. Therefore, we have a true polynomial time approximation algorithm for fair clustering with approximation ratio at most $\alpha(\mathcal{I})$.

Now we prove the second part. By definition the output of a true approximation for fair clustering would have no proportional violations therefore achieving the optimal value for any objective for **FCBC**. Therefore, we have an optimal algorithm for **FCBC** for $U \geq \alpha'(\mathcal{I}) \,\mathrm{OPT}_{FC}(\mathcal{I}) \geq \alpha'(\mathcal{I}) \,\mathrm{OPT}_{cb}(\mathcal{I})$. $\quad\square$

Further, a true approximation algorithm for fair clustering algorithm would imply an exact algorithm for fair clustering under a bounded cost **FCBC**.

**Theorem A.1.** *A true polynomial time $\alpha(\mathcal{I})$ approximation algorithm for fair clustering implies that fair clustering under a bounded cost* **FCBC** *can be solved optimally in polynomial time for $U \geq \alpha(\mathcal{I}) \,\mathrm{OPT}_{FC}(\mathcal{I})$.*

*Proof.* Since we have a true approximation, then there would be no fairness violation (optimal value). Further, this would be $U$ such that $U \geq \alpha(\mathcal{I}) \operatorname{OPT}_{\mathbf{FC}}(\mathcal{I}) \geq \alpha(\mathcal{I}) \operatorname{OPT}(\mathcal{I})$. $\qquad\square$

## B Solution for the GROUP-LEXIMIN Case

Here we present the full details for the GROUP-LEXIMIN case, see algorithm (5) below.

---

**Algorithm 5** Leximin Algorithm

---

**Input:** set of points $\mathcal{C}$, price of fairness $U$, for each color $h \in \mathcal{H}$ lower and upper proportion values $\beta_h, \alpha_h$
$\mathcal{H}_{\text{fixed}} = \{\}, \Delta_{\min} = 0$
**while** $\mathcal{H}_{\text{fixed}} \neq \mathcal{H}$ **do**
$\quad$**Step 0:** For each color $h \in \mathcal{H}_{\text{fixed}}$ set its violation to its minimum found in the previous iterations $\Delta_h^{\min}$.
$\quad$**Step 1:** For each color $h \in \mathcal{H} - \mathcal{H}_{\text{fixed}}$, set $\Delta_h = \Delta$.
$\quad$Find the the minimum $\Delta$, such that $\Delta < \Delta_{\min}^{q-1}$ that satisfies LP (7) using binary search over $E_\epsilon$, let $\Delta_{\min}^q = \Delta + \frac{2}{L}$.
$\quad$**Step 2:** For the set $h'_\ell \in \mathcal{H} - \mathcal{H}_{\text{fixed}}$, find the minimum set of colors with violation $\Delta_{\min}^q$ and add them to $\mathcal{H}_{\text{fixed}}$ using LP (9).
$\quad$**Step 3:** If in **Step 2** no color is found, then randomly pick a color from $\mathcal{H} - \mathcal{H}_{\text{fixed}}$ and add it to $\mathcal{H}_{\text{fixed}}$.
**end while**

---

LP (9) below is run once for each $h'_\ell \in \mathcal{H} - \mathcal{H}_{\text{fixed}}$ (note lines 9c and 9e). The LP does not have an objective and amounts to a feasibility check.

$$\sum_{i,j} d^p(i,j) x_{ij} \leq U^p \tag{9a}$$

$$\forall j \in \mathcal{C} : \sum_{i \in S} x_{ij} = 1, \quad x_{ij} \in [0,1] \tag{9b}$$

$$\tag{9c}$$

for the given $h'_\ell$:

$$\left(\beta_h - \left[\Delta_{\min}^q - \frac{2}{L} - \epsilon\right]\right)\left(\sum_{j \in \mathcal{C}} x_{ij}\right) \leq \sum_{\substack{j \in \mathcal{C}, \\ \chi(j) = h'_h}} x_{ij} \leq \left(\alpha_h + \left[\Delta_{\min}^q - \frac{2}{L} - \epsilon\right]\right)\left(\sum_{j \in \mathcal{C}} x_{ij}\right) \tag{9d}$$

$$\forall h \in \mathcal{H} - \left(\mathcal{H}_{\text{fixed}} \cup \{h'_\ell\}\right), \forall i \in \{1, \ldots, k\}: \tag{9e}$$

$$\left(\beta_h - \Delta_{\min}^q\right)\left(\sum_{j \in \mathcal{C}} x_{ij}\right) \leq \sum_{\substack{j \in \mathcal{C}, \\ \chi(j) = h}} x_{ij} \leq \left(\alpha_h + \Delta_{\min}^q\right)\left(\sum_{j \in \mathcal{C}} x_{ij}\right) \tag{9f}$$

$$\forall h \in \mathcal{H}_{\text{fixed}} : \left(\beta_h - \Delta_h^{\min}\right)\left(\sum_{j \in \mathcal{C}} x_{ij}\right) \leq \sum_{\substack{j \in \mathcal{C}, \\ \chi(j) = h}} x_{ij} \leq \left(\alpha_h + \Delta_h^{\min}\right)\left(\sum_{j \in \mathcal{C}} x_{ij}\right) \tag{9g}$$

In algorithm (5), step 1 does a binary search over the set $E_\epsilon$ to find the minimum feasible violation $\Delta_{\min}^q$ for the active set of colors (in the set $\mathcal{H} - \mathcal{H}_{\text{fixed}}$). Step 2 finds the set of colors whose violation can be improved beyond $\Delta_{\min}^q$, by running an LP (see B) specific to each color in $\mathcal{H} - \mathcal{H}_{\text{fixed}}$. If all colors in $\mathcal{H} - \mathcal{H}_{\text{fixed}}$ can improve, then in step 3 a random color is picked from that a set. Step 0 simply sets the violation to the optimal found value for the set of colors that can no longer be improved (the set $\mathcal{H}_{\text{fixed}}$).

# C Discussion on the Size of the Smallest Cluster for a Given Cost Upper Bound

As discussed, it is clear from Theorem 6.1 that the larger the size of the smallest cluster, the better our approximation. In Section C.1 we consider examples where in the absence of outliers and for suitable values of $k$, we would not have small clusters. We note further that whereas the given cost is $U$ the final approximation in Theorem 6.1 is in terms of $\frac{1}{L(U')}$ where $U' = (2 + \alpha)U$ with $\alpha$ being the color-blind approximation ratio. So it is reasonable to wonder about the gap between $L(U)$ from the lower bound of Theorem 5.3 and $L(U')$ from Theorem 6.1. We show that the gap is example dependent, specifically in Section C.2 we show that the gap can be arbitrarily large and in Section C.3 we show that it's possible that they are precisely equal, i.e. $L(U) = L(U')$. We note here that color assignments have no significance.

## C.1 For suitable $k$ with no outliers $L(U)$ is large

For reasonable values of $k$ and in the absence of outliers, provided the upper clustering cost $U$ is not very large, we do not expect clusters of a small size, i.e. $L(U)$ is large. In Figure 8 we choose a value of $k$ that recovers the underlying structure. We can see that there are no small clusters. On the other had, in Figure 9 because of outliers we end up with small clusters. Moreover, if we choose a value of $k$ greater than 3, then we notice of the clusters fragment as in Figure 10, this leads to smaller clusters but even then they are not pathologically small.

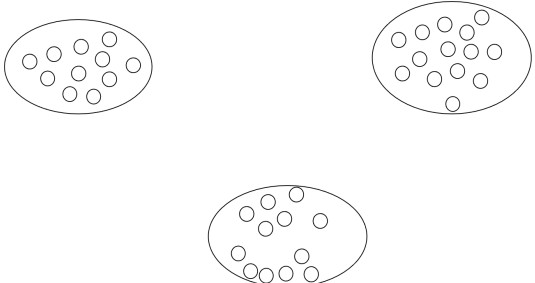

Figure 8: Here there are no outliers and we choose $k = 3$ which recovers the clusters of the dataset.

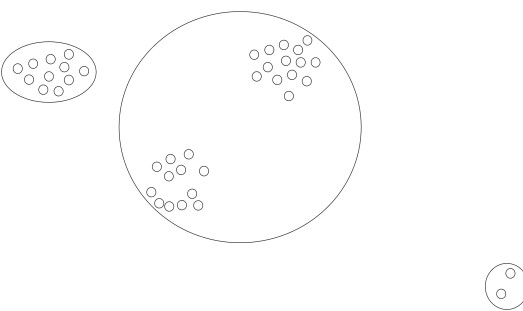

Figure 9: Because of outlier points, the smallest cluster consists of only two points.

## C.2 Example where the gap between $L(U)$ and $L(U')$ is large

See Figure 11 which is the same as that of Figure 4. Note that by Lemma A.4, for the $k$-center if $U = 1$, then $L(U) = 4$. But at $U' = (2 + \alpha)U = 4$, we can get $L(U') = 1$. This is not difficult to see since if we select any point from the top row $\mathcal{T}$, then every other point in the graph is at a distance of at most 2.

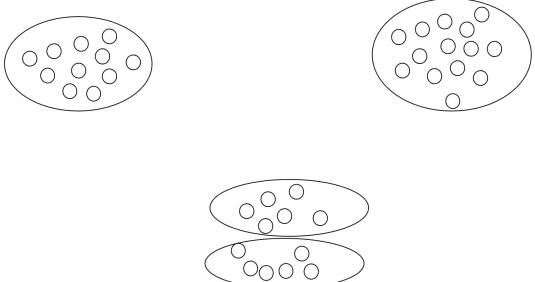

Figure 10: Although it can be argued that $k = 3$ is more suitable for this example. Setting $k = 4$ does not lead to very small clusters.

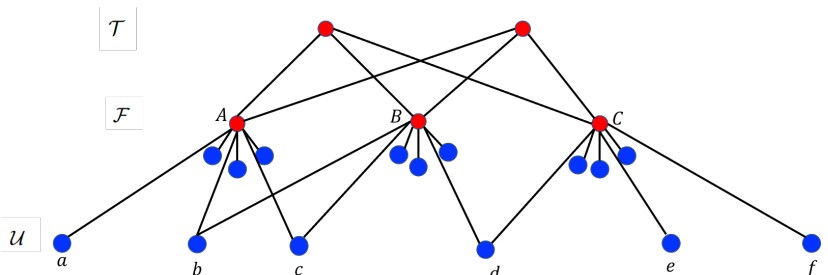

Figure 11: This is an example where the change in the size of the smallest cluster is very large. At $U = 1$, we have $L(U) = 4$. However, at $U' = 4$, we have $L(U') = 1$.

## C.3  Example where $L(U') = L(U)$

See Figure 12 with $k = 3$ and the distance between the clusters $R$ is sufficiently large, the size of the smallest cluster does not decrease with increasing cost for reasonable values of $U$. For the $k$-center, if we have $U = r$, then $U' = (\alpha + 2)r$ where $r$ is the maximum radius of the "true" clusters. If $R > (2 + \alpha)r$ then we would not have small clusters.

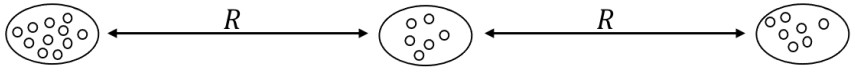

Figure 12: In this example because of the large separation between the clusters, we would have $L(U) = L(U')$ for reasonable values of $U$.

## D  Tradeoff between the Clustering Cost Bound and Fairness

It is worth asking if we can obtain a clear tradeoff between the upper clustering cost bound and the achievable fairness (whether using the GROUP-EGALITARIAN or GROUP-UTILITARIAN objectives). In the two sections bellow, we show that in general this is not achievable. In some examples, the tradeoff is effectively a step function whereas in others it gradually increases. Throughout, we consider two clustering ($k = 2$) with the $k$-center objective and set $\alpha_{\text{red}} = \alpha_{\text{blue}} = \beta_{\text{red}} = \beta_{\text{blue}} = \frac{1}{2}$. By the above proportions, it is not difficult to see that $\Delta_{\text{red}} = \Delta_{\text{blue}}$ and that GROUP-UTILITARIAN = 2GROUP-EGALITARIAN, simply following the proofs of Lemma A.5 and Lemma A.6, respectively. Since GROUP-UTILITARIAN = 2GROUP-EGALITARIAN, we will only discuss the GROUP-EGALITARIAN objective for simplicity. Further, it follows as well that $\Delta_{\text{red}}, \Delta_{\text{blue}} \leq \frac{1}{2}$ and that any non-trivial color-proportional clustering should have $\Delta_{\text{red}}, \Delta_{\text{blue}} < \frac{1}{2}$.

### D.1 Example where the tradeoff is a step function

In the following examples (a) and (b) shown in Figure 13 the tradeoff follows a step function. Letting $r$ be the distance between the nearby same color points and letting $R$ be the smallest distance between points of different color. It is clear that if $U < R$, then GROUP-EGALITARIAN $= \frac{1}{2}$. However, if $U \geq R$, then GROUP-EGALITARIAN $= 0$. Note that we can make the value of $R$ arbitrarily large as shown in (a) and (b).

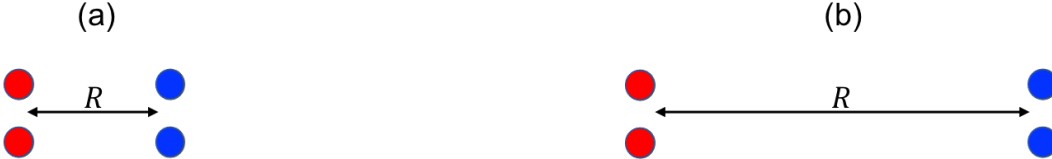

Figure 13: Examples where the tradeoff between the upper cost bound and the fairness objective is abrupt.

### D.2 Example where the fairness increases gradually with higher cost upper bound

Consider the case where the points lie on a line and $|\mathcal{C}^{\text{red}}| = |\mathcal{C}^{\text{blue}}| = \frac{|\mathcal{C}|}{2} = n'$ where $n'$ is odd. Clearly, $|\mathcal{C}| = n = 2n'$. Further, let all of the first $n'$ points be red and the reaming $n'$ points be blue. The distance between any two consecutive points is $r$. Figure 14 shows an example of such a construction for $n' = 5$.

It is not difficult to see that for the two clustering case ($k = 2$), that the optimal radius is $(\frac{n'-1}{2})r$, simply set the middle point of each color as the center. Note that we would have GROUP-EGALITARIAN $= \frac{1}{2}$, i.e. no mixing of the colors.

If we index the upper $U$ by the following $U = U(m) = (\frac{n'-1}{2} + m)r$ where $m$ is an non-negative integer and $m < \frac{n'}{4}$, then it is not difficult to show that the achievable fairness is GROUP-EGALITARIAN$(m) = \frac{1}{2} - \frac{2m}{n'}$. That is, if we allow the cost to increase by $mr$, then we can improve the GROUP-EGALITARIAN objective by $\frac{2m}{n'}$.

For example, in Figure 14, for $U = U(0) = 2r$, then we cannot mix the color and GROUP-EGALITARIAN $= \frac{1}{2}$. If on the other hand, we choose $U = U(1) = 3r$, then for the red color we can have a representation of $\frac{2}{5}$, leading to GROUP-EGALITARIAN$(1) = \frac{1}{2} - \frac{2}{5}$.

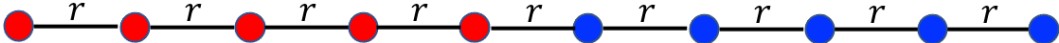

Figure 14: In the above example the tradeoff between the clustering cost and the fairness objective is gradual.

## E Network Flow Rounding

Here, we summarize the network flow rounding due to [11] as applied to our problem. Recall that it gives the following guarantees:

(i) $\sum_{i,j} d^p(i,j)\bar{x}_{ij} \leq \sum_{i,j} d^p(i,j)x_{ij}$.

(ii) $\forall i \in \{1, \ldots, k\} : \left\lfloor \sum_{j \in \mathcal{C}} x_{ij} \right\rfloor \leq \sum_{j \in \mathcal{C}} \bar{x}_{ij} \leq \left\lceil \sum_{j \in \mathcal{C}} x_{ij} \right\rceil$

(iii) $\forall h \in \mathcal{H}, \forall i \in \{1, \ldots, k\} : \left\lfloor \sum_{j \in \mathcal{C}^h} x_{ij} \right\rfloor \leq \sum_{j \in \mathcal{C}^h} \bar{x}_{ij} \leq \left\lceil \sum_{j \in \mathcal{C}^h} x_{ij} \right\rceil$

The first guarantee essentially states that we can round the fractional LP assignments to an integral solution without increasing the cost given that we solved the LP for a fixed set of centers. However, proving this depends on the objective. For $k$-center, assigning a point $j$ to any center $i$ with $x_{ij} > 0$

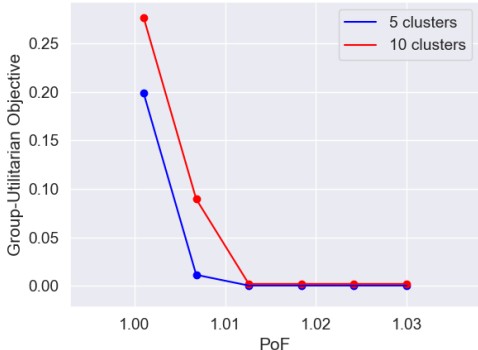

*Figure 15:* PoF *versus the* GROUP-UTILITARIAN *objective for the **CreditCard** dataset with* $\delta = 0.05$.

will not affect the cost of the solution with respect to the clustering objective. This is because $k$-center only minimizes the maximum radius of any cluster and we only allow $x_{ij} > 0$ if $d^p(i, j)$ is at most the maximum radius. On the other hand, a fractional solution to the LP for $k$-median or $k$-means may fractionally assign part of a point to a nearby center and part to a faraway center to improve fairness. A rounded integral solution that assign the point wholly to the farther center could potentially increase the cost. The proof due to [11] defines these objectives as reassignable ($k$-center) and separable ($k$-median and $k$-means) and focuses on the more challenging case of separable objectives.

We create a min-cost flow instance with unit capacities to solve this problem as follows.

- For each center $i \in S$, we define a subset of points $V_i$ with a point $v_i^h$ for each color $h \in \mathcal{H}$ and a main point $v_i^0$. Each $v_i^h$ has a balance of $-\left\lfloor \sum_{j \in \mathcal{C}^h} x_{ij} \right\rfloor$ and $v_i^0$ has a balance of $-(\left\lfloor \sum_{j \in \mathcal{C}} x_{ij} \right\rfloor - \sum \left\lfloor \sum_{j \in \mathcal{C}^h} x_{ij} \right\rfloor)$. The arc set of these points is $(v_i^h, v_i^0)$ for each color $h \in \mathcal{H}$ and each arc has cost 0.

- For each point $j \in \mathcal{C}$, we define a point $v_j$ with balance of 1. For each $x_{ij} > 0$ we add an arc $(v_j, v_i^h)$ with cost $d^p(i, j)$ where $h$ is the color of $j$.

- Finally, we add a sink $t$ with balance $-(|\mathcal{C}| - \sum i \in S \left\lfloor \sum_{j \in \mathcal{C}} x_{ij} \right\rfloor)$. For each center $i$, we add an arc $(v_i^0, t)$ with cost 0.

Note that all of the capacities, costs, and balances are integral and that the LP solution translates to a feasible flow. Thus, we can find an integral flow solution with cost at most that of LP solution and it is easy to see that this can then be translated back to an integral assignment. Also, note that our flow solution almost preserves the fairness of the LP solution. The additive error of 1 for the second and third guarantees above arise from taking the floor (e.g., $\left\lfloor \sum_{j \in \mathcal{C}^h} x_{ij} \right\rfloor$) to have integrality.

# F Additional Experimental Results

In this section, we report additional experimental result on the $k$-means and $k$-median objectives. We also use the **CreditCard** dataset from the UCI repository [24] with all 30,000 data points. The fairness attribute we use is marriage where we merge groups 2 (single) and 3 (other) into one group to have a binary attribute.

## F.1 Additional $k$-means Results

### F.1.1 GROUP-UTILITARIAN Objective

Here we add results for the GROUP-UTILITARIAN for the **CreditCard** dataset with $\delta = 0.05$, see Figure 15. We see that we can achieve better proportional violations for the smaller value of $k$.

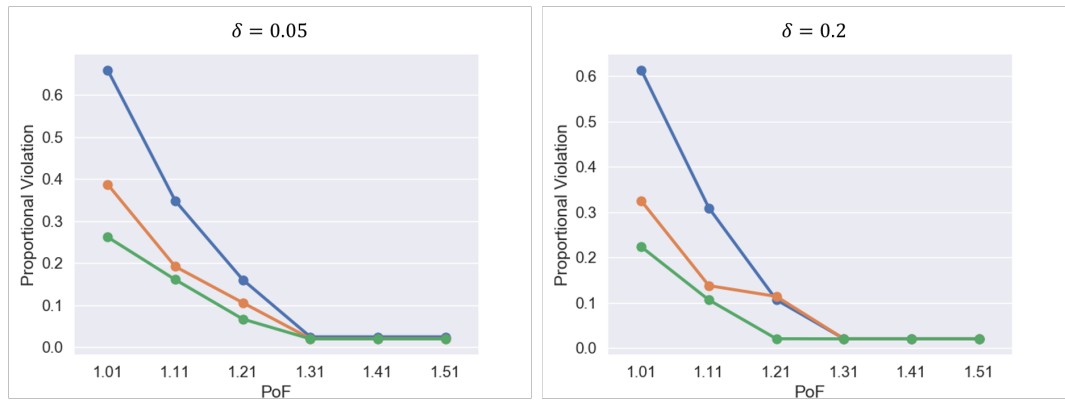

*Figure 16:* PoF *versus the* GROUP-LEXIMIN *objective for the* ***Census1990*** *dataset for different values of δ.*

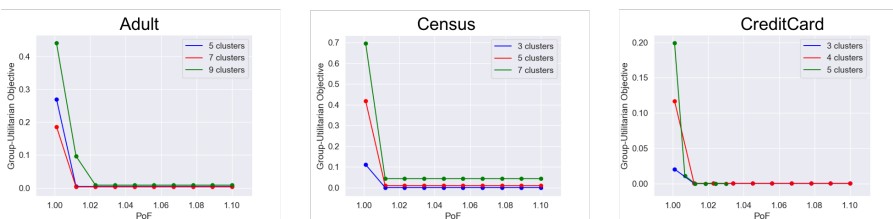

*Figure 17:* PoF *versus the* GROUP-UTILITARIAN *objective for the* ***Adult***, ***Census1990***, *and* ***CreditCard*** *datasets with for the* $k$-*median objective.*

### F.1.2 GROUP-LEXIMIN Objective

Figure 16 shows the results for GROUP-LEXIMIN on the **Census1990** dataset where we have $k = 5$. We set $\delta = 0.05$ and $\delta = 0.2$, we see similar behaviour to that in Figure 3. We also notice that for higher values of $\delta$ (more relaxed proportion bounds) some colors are able to achieve smaller proportional violation which we expect.

### F.2 $k$-median Results

For the color-blind implementation of the $k$-median objective we follow [10, 22] and use the 5-approximation of [6] with modified $D$-sampling [5]. Because we are interested to see the behaviour of the algorithm and since the color-blind approximation is time consuming we sub-sample all datasets to 1,000 points.

### F.2.1 GROUP-UTILITARIAN Objective

Figure 17 shows the performance on the different datasets for the GROUP-UTILITARIAN objective. Note that we observe a similar trend as in figure 2 where it is easier to minimize the proportional violations when the number of clusters is lower. Note that we set $\delta = 0.1$ for all datasets.

### F.2.2 GROUP-LEXIMIN Objective

Figure 18 shows the results on the GROUP-LEXIMIN objective on the **Census1990** dataset with $k = 5$ and $\delta = 0.1$. The behaviour is very similar to the $k$-means.

### F.3 Checking the Size of the Smallest Cluster

Here we check the size of the smallest cluster again, for Section F.1 the smallest cluster is of size 176 for the **CreditCard** dataset and 168 for the **Census1990** dataset.

In Section F.2 where the datasets have been sub-sampled to 1,000 points. The smallest cluster size found is 27, 15, and 24 for the **Adult**, **Census1990**, and **CreditCard** datasets, respectively.

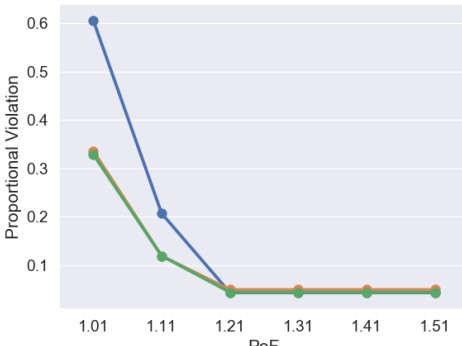

*Figure 18:* PoF *versus the* GROUP-LEXIMIN *objective for the **Census1990** dataset with $k = 5$ and $\delta = 0.1$ for the $k$-median objective.*

It is clear from these experiments that size of the smallest cluster is large and therefore the approximations we obtain are not far from the optimal.

## G   Details of the Randomized Extension

We give a sketch of the algorithm, and defer some of the proof and details to the full version. Suppose the LP returns a solution $x_{i,j} \in [0,1]$ for all $(i,j)$, satisfying the following equalities for some non-negative quantities $a_i$ and $b_{i,h}$:

1. $\forall j \in \mathcal{C}: \sum_i x_{ij} = 1$.
2. $\forall i \in \{1,\ldots,k\} : \sum_{j \in \mathcal{C}} x_{ij} = a_i$.
3. $\forall h \in \mathcal{H}, \forall i \in \{1,\ldots,k\} : \sum_{j \in \mathcal{C}^h} x_{ij} = b_{i,h}$.
4. $\forall (i,j) : x_{i,j} \in [0,1]$.

We will sketch a randomized algorithm that gradually rounds the $x_{i,j}$'s to the eventual $\bar{x}_{i,j}$ in a polynomial number of iterations. Let $Y_{i,j,t}$ denote the (random) value of $x_{i,j}$ at the end of iteration $t$; initially, we deterministically have $Y_{i,j,0} = x_{i,j}$. Let $A_{i,t}$ and $B_{i,h,t}$ be random variables such that

- $\forall i \in \{1,\ldots,k\} : A_{i,t} = \sum_{j \in \mathcal{C}} Y_{i,j,t}$.
- $\forall h \in \mathcal{H}, \forall i \in \{1,\ldots,k\} : B_{i,h,t} = \sum_{j \in \mathcal{C}^h} Y_{i,j,t}$.

Let us now describe iteration $t \geq 1$, which operates on the values $Y_{i,j,t-1}$ and probabilistically modifies them to the $Y_{i,j,t}$. Fix the values $Y_{i,j,t-1}$ to be some arbitrary $y_{i,j,t-1}$, and define $a_{i,t-1} = A_{i,t-1}$, $b_{i,h,t-1} = B_{i,h,t-1}$. We will maintain the following **five invariants**:

**(I1)** $\forall j \in \mathcal{C}: \sum_i Y_{i,j,t} = 1$ with probability one.

**(I2)** $\forall i \in \{1,\ldots,k\} : \lfloor a_{i,t-1} \rfloor \leq A_{i,t} \leq \lceil a_{i,t-1} \rceil$ with probability one.

**(I3)** $\forall h \in \mathcal{H}, \forall i \in \{1,\ldots,k\} : \lfloor b_{i,h,t-1} \rfloor \leq B_{i,h,t} \leq \lceil b_{i,h,t-1} \rceil$ with probability one.

**(I4)** $E[Y_{i,j,t}] = y_{i,j,t-1}$.

**(I5)** $Y_{i,j,t} \in [0,1]$ with probability one.

In particular, we have the following key properties:

- if $a_{i,t-1}$ is an integer, then $A_{i,t} = a_{i,t-1}$ with probability one;
- if $b_{i,h,t-1}$ is an integer, then $B_{i,h,t} = b_{i,h,t-1}$ with probability one; and
- if $y_{i,j,t-1}$ is an integer (which will be 0 or 1), then $Y_{i,j,t} = y_{i,j,t-1}$ with probability one.

Our strategy is to show that there is a way of maintaining our invariants above, while making at least one *more* $A_{i,t}$, $B_{i,h,t}$, or $Y_{i,j,t}$ integral at the end of iteration $t$. Since there is only a polynomial number of these terms and since we are done when all the $Y_{i,j,t}$'s are integers, our proof of correctness will then be complete by a simple induction on $t$.

Briefly, iteration $t$ starts with the following constraint system with variables $z_{i,j}$, initialized with the feasible solution $z_{i,j} = y_{i,j,t-1}$:

**(C1)** $\forall j \in \mathcal{C} : \sum_i z_{ij} = 1$.

**(C2)** $\forall i \in \{1, \ldots, k\}$ *such that* $a_{i,t-1}$ *is an integer* $: \sum_{j \in \mathcal{C}} z_{ij} = a_{i,t-1}$.

**(C3)** $\forall h \in \mathcal{H}, \forall i \in \{1, \ldots, k\}$ *such that* $b_{i,h,t-1}$ *is an integer* $: \sum_{j \in \mathcal{C}^h} z_{ij} = b_{i,h,t-1}$.

**(C4)** $\forall (i, j)$ *such that* $y_{i,j,t-1}$ *is an integer (i.e., lies in* $\{0, 1\}$) $: z_{i,j} = y_{i,j,t-1}$.

Given our initialization of $z$, we also have that $\forall (i, j) : z_{i,j} \geq 0$.

We prove below via a careful counting argument that the system (C1)-(C4) of the form $Az = v$ is an under-determined system. Thus, there is a nonzero vector $r$ that is efficiently computable such that $Ar = 0$. We then calculate certain positive scalars $u_1$ and $u_2$, and probabilistically transition from the vector $y = (y_{i,j,t-1})$ to the random vector $Y = (Y_{i,j,t})$ as follows:

- with probability $u_2/(u_1 + u_2)$, set $Y = y + u_1 r$;
- with the remaining probability of $u_1/(u_1 + u_2)$, set $Y = y - u_2 r$.

Briefly, $u_1$ and $u_2$ are chosen positive and just large enough so that we maintain our five invariants, while making at least one *more* $A_{i,t}$, $B_{i,h,t}$, or $Y_{i,j,t}$ integral at the end of iteration $t$.

**Proof that (C1)-(C4) is an under-determined system.** Let us call pair $(i, j)$ *rounded* if $y_{i,j,t-1}$ lies in $\{0, 1\}$, and *floating* otherwise (i.e., if $y_{i,j,t-1}$ lies in $(0, 1)$). Let $R$ and $F$ respectively denote the sets of rounded and floating pairs.

We next remove two types of redundant constraints from our system (C1)-(C4):

**(R1)** Given the constraints (C4), we can remove any constraint in (C1), (C2), or (C3) in which all pairs $(i, j)$ that appear in the LHS of the constraint, lie in $R$: such constraints are redundant given (C4), and are removed.

**(R2)** If the constraint for $i$ in (C2)—if it appears in (C2)—is linearly dependent on the constraints for $(i, \cdot)$ in (C3),[7] then we say that $i$ *is (C2)-redundant* and remove the constraint for $i$ in (C2).

Clearly, removal of these redundant equalities does not change our system. *From now on, (C1)-(C4) refers to the reduced system **after** the removal of these redundant constraints.*

We next define certain numbers of floating indices after the removals in (R1) and (R2). If the constraint for $j$ appears in (C1), let $N_1(j)$ denote the number of floating pairs $(\cdot, j)$ in the LHS of this constraint. The second number of floating indices is defined in a slightly-more-refined manner: if the constraint for $i$ appears in (C2), let $N_2(i)$ denote the number of floating pairs $(i, \cdot)$ in the LHS of this constraint, *that do not appear in the LHS of any of the constraints for $(i, \cdot)$ in (C3)*. Next, if the constraint for $(i, h)$ appears in (C3), let $N_3(i, h)$ denote the number of floating pairs $(i, \cdot)$ in the LHS of this constraint.

We next make three useful observations. Note first that by (R1), we have $N_1(j)$ and $N_3(i, h)$ are positive; in fact, since the constants in the RHS of (C1) and and (C3) are all *integers*, a moment's thought reveals that each of $N_1(j)$ and $N_3(i, h)$ is *at least two*. Second, we claim that each $N_2(i)$ is at least two as well. Indeed, since the constraint for $i$ appears in (C2), we must have that this constraint is linearly independent of (in particular, is not the sum of) the constraints for $(i, \cdot)$ in (C3); hence, the LHS of the constraint for $i$ in (C2) must have at least one variable not covered by the constraints for $(i, \cdot)$ in (C3), implying that $N_2(i)$ is positive as well. The fact that the constants in the RHS of (C2) are integers, again implies that $N_2(i) \geq 2$. Third, suppose $a_{i,t-1}$ is *not* an integer,

---

[7]This simply means here that the constraint for $i$ in (C2) is the sum of the constraints for $(i, \cdot)$ in (C3), since the latter are all supported on pairwise-disjoint sets of variables.

in which case we will say $i$ is "(C2)-not-integral." In this case, it is easy to see that there must exist some $h$ such that $z_{i,h}$ *does not* appear in the LHS of any of the constraints (C2) and (C3).

We are finally to ready to prove that (C1)-(C4) is underdetermined. Note that the number of constraints in (C4) is $|R|$, and that the total number of variables is $|R| + |F|$. Let $n_1$, $n_2$, and $n_3$ denote the respective numbers of constraints in (C1), (C2), and (C3). Recall again that these (C1)-(C3) refer to the system after applying the removal steps (R1) and (R2).

Since each variable in (C1) appears only once in (C1) and since $N_1(j) \geq 2$, we get that

$$|F| \geq 2n_1. \tag{10}$$

Similarly, from the consideration of (C2) and (C3) along with the observation above about indices $i$ that are (C2)-not-integral, we get

$$|F| \geq 2(n_2 + n_3) + 1 \text{ if there is a (C2)-not-integral } i \tag{11}$$
$$\geq 2(n_2 + n_3) \text{ otherwise} \tag{12}$$

Thus, by averaging the above and using the fact that $|F|$ is an integer, we obtain

$$|F| \geq n_1 + n_2 + n_3 + 1 \text{ if there is a (C2)-not-integral } i \tag{13}$$
$$\geq n_1 + n_2 + n_3 \text{ otherwise} \tag{14}$$

Now, if our system is *not* under-determined, we must have that the total number of constraints, which is $n_1 + n_2 + n_3 + |R|$, is at least as large as the total number of variables $|F| + |R|$. We see from (13) that this is impossible if there is a (C2)-not-integral $i$. Thus we may assume that there is **no** (C2)-not-integral $i$, and hence that the number of constraints exactly equals the number of variables. We will next show that this system is linearly dependent, which, from the fact that the number of constraints equals the number of variables, will show that the system is under-determined. Assume for convenience that no constraint in (C2) for any $i$ was removed in (R2) (if not, we simply replace this constraint by the sum of the constraints for $(i, \cdot)$ in (C3), in the following argument). Then, a moment's reflection shows that the sum of the constraints in (C1) equals the sum of the constraints in (C2), yielding the desired linear dependence.