# OpenReview forum: "Fair Clustering Under a Bounded Cost"
_NeurIPS.cc/2021/Conference — NeurIPS 2021 Poster_

### Official Review · Reviewer_aWDm · 2021-06-28

**Rating:** 6
**Confidence:** 4

**Summary:**

This paper derives the theoretical bound of fair clustering based on porotype based algorithms. Generally speaking, the authors expect to minimize the unfairness under the given upper bound. In this paper, the authors consider two types of fairness and two fairness clustering setting. Based on this, the authors derive the upper bound of the unfairness and design the corresponding algorithm. The major contributions of this paper lie in theoretical analysis.



**Ethical Concerns:**

I do not have any ethical concerns on this paper.

**Limitations And Societal Impact:**

In the checklist, the authors confirm that they include the limitations and societal impact in the main paper. I do not find them in both main paper and supplementary document.

**Main Review:**

Here are my detailed comments.

1. The authors expect to provide a comprehensive understanding on the research problem with different objective functions and different porotype clustering algorithms. In the current version, there are two research problems and three objective functions. The authors elaborate the whole paper in a parallel way. Unfortunately, it makes this paper messy and difficult to follow. I suggest the authors to only focus on the simplest scenario, and first provide the complete procedure from problem definition to bound cost to algorithm. Then, other scenarios can be delivered as extensions.
2. Figure 1 is not easy to understand. I read this part for more than three times. Maybe only the bottom example?
3. I was wondering for several days whether the problem addressed in this paper is meaningful. In Paper "Fair Algorithms for Clustering", they minimize the clustering cost with fairness constraints, while the authors of this paper formulate the problem to minimize the unfairness under the clustering cost. It seems that these two settings should achieve the similar results. However, the problem in this paper might impractical and even inflexible. I have two points on this.  (a) cluster analysis is an unsupervised task, the clustering cost is not a focused measurement due to its unachievable OPT; therefore it is difficult to set the upper bound. The authors introduce the concept of PoF, where k-means++ is used as a proxy of the lowest possible value of the clustering cost. What does PoF=1.1 mean in practical? (b) Although the authors introduce \delta_h to relax the original optimization problems in Eq. (1) or (2), there still exists the empty solution. In light of this, I would like to support the setting that minimizes the clustering cost with fairness constraints, rather than the one of this paper. For this point, I need to discuss with other reviewers.
4. The presentation is a little messy. Theorem 6.1 talks about Algorithm 1, while Algorithm 1 comes behind Theorem 6.1.
5. In Line 197, what does the "small" mean?
6. I am confused on ALG-FCBC. According to its formulation, the set of centers S are not fixed, which should be optimized during the clustering procedure. However, ALG-FCBC first calls a color-blind clustering algorithm to obtain S, and keeps S unchanged.
7. The essence of ALG-FABC is multiple LPs. In my eyes, the solution is almost brute-force. The authors do not provide the time complexity analysis.
8. This paper focuses on theoretical analysis. Moreover, these theorems cannot be verified by experiments. The experimental section can be removed. Instead, a conclusive section is needed.

With the below discussion with authors, my major concern on problem setting is addressed and I would like to increase my score.

**Time Spent Reviewing:**

24

---

> ### Author Response · Authors · 2021-08-10
> **Response to Reviewer4(aWDm)**
>
> We thank the reviewer for the detailed comments. Here we provide our response for each point:
>
> 1-$\textbf{Having Many Objectives and Problems}$: While it may be more clear to provide a solution for a specific objective and constraint, this would also come at the expense of highlighting the generality of our algorithms and how they follow a similar approach but with different subroutines for different objectives and constraints. Further, while the objectives are different they are closely related (e.g. GROUP-UTILITARIAN $\ge$ GROUP-EGALITARIAN), and the clustering cost constraints simply involve a different value for $p$ (e.g., $p=1$ and $p=2$ for the $k$-median and $k$-means, respectively). Moreover, such an approach was followed before in the literature; see references [$\alpha,\beta,\gamma$].
>
> 2-$\textbf{Figure 1 Issue}$: In Figure 1, each cell is a clustering example where color-blind clustering is applied on the right and fair clustering is applied on the left. The examples in the same row are actually identical. Our point is that the increase in the clustering cost that is incurred by imposing fairness is example dependent. In the upper row example, it is not high. However, in the lower example it is much higher (where the clusters are containing highly separated points). One can also see that the increase in the cost to achieve fairness is actually unbounded (by keeping the same color points at the same distance from each other but separating them further from the different color points). We will add a more detailed caption to the figure and also enlarge its size to clarify this point.
>
> 3-$\textbf{Whether the Problem is Meaningful}$: First, we note that although clustering problems are NP-hard in the worst case, there are many approximation algorithms for them. Furthermore, in practice, even heuristics such as Lloyd's algorithm are known to perform quite well, i.e., achieve near-optimal clusterings.
>
> $\textbf{(a)}$:The reviewer’s issue is with the motivation. To motivate the problem further, we can elaborate on our discussion in the introduction. Suppose a practitioner clusters a given dataset and obtains a cost $c_1$ (which is not necessarily the optimal cost) but finds it to be highly unfair, i.e., the color proportions in the clusters are highly un-representative of the population. To achieve fairness, the practitioner instead applies a fair clustering algorithm and obtains a cost $c_2$. If they then find that $c_2$ is much larger than $c_1$, then it follows that the clustering quality has been degraded which means that farway points have likely been put in the same clusters (as shown in the bottom left example in Figure 1). Some visualization methods could even show that.
>
> Now, if the practitioner applies this low quality fair clustering for a task such as recommendation/market segmentation, the output despite being fair is likely to be of low value. Therefore, since the legal notion of disparate impact allows the violation of group fairness if it can be shown to come at a high expense (lines 50-52), the practitioner is likely to forgo the fair algorithm and instead use the color-blind (“unfair”) algorithm. However, the practitioner may decide that he can minimize unfairness at the expense of “loosing” some quality to the clustering by setting some upper bound value $c_3$, where $c_3$ is larger than $c_1$ by some factor (this can be better chosen by knowing the application and/or trial and error). Accordingly, whether a PoF=1.1 is acceptable or not is decided by the practitioner who does clustering and has insight into its effects on downstream applications.
>
> Our paper addresses this situation by allowing the practitioner to set the upper bound $c_3$ that they consider acceptable and achieving as much fairness as possible (minimizing unfairness) instead.
>
>
>
>
> $\textbf{(b)}$: For this point, we would appreciate clarification from the reviewer’s side during the discussion phase.  Specifically, we ask the reviewer to clarify what is meant by the empty solution still existing in Eq(1) and Eq(2)? Here, if empty solution means an empty cluster, i.e., $|\mathcal{C}_i|=0$, then this is actually allowed and is not new in our work. We also have $|S|\leq k$ instead of $|S| = k$. In fact, it is necessary to allow empty clusters. To see that, consider doing fair clustering over a set of 5 blue points and 7 red points and that the proportion bounds require precisely a 5:7 (blue:red) representation in each cluster. Since 5 and 7 are relatively prime, the only feasible way to do that is to cluster all of the points in one cluster. Therefore, if we insist on having more than one cluster the problem would be infeasible. Furthermore, the above example can be generalized to a larger number of points, larger number of clusters, and relaxed proportion bounds.
>
> 4-$\textbf{Algorithm 1 comes behind Theorem 6.1}$: Unfortunately, due to the page limit we had to find ways to fit the results. We will make our best attempt to improve this detail of the presentation, potentially making use of additional pages should that option arise.
>
>
>
> 5-$\textbf{Line 197, what does the ``small" mean?}$ Small clusters mean clusters of a small size, i.e. including a small number of points. If small clusters do not exist for the given value of $U$, then it follows that $L(U)$ is larger. This implies that the approximations lower bounds (Theorem 5.3) and the approximations that we achieve (Theorem 6.1) are smaller (more accurate) since they depend on the reciprocal of the smallest cluster size.
>
> 6-$\textbf{The set of centers $S$ is found in ALG-FCBC and then fixed}$: It is correct that in ALG_FCBC a color-blind clustering algorithm is called to find $S$, and that $S$ remains unchanged; however, in the proof of Theorem 6.2 we show that we can still establish bounds on the clustering cost and unfairness using this choice of centers.
>
> 7-$\textbf{The solution appears to be a brute force and there is no time complexity analysis}$: Although the algorithm runs multiple LPs, we show that the number has a logarithmic dependence on $1 / \epsilon$ (Theorems 6.5 and 6.6). Moreover, we provide a “query” complexity on the number of LPs we need to run. When LPs are used, it is common not to write the full time dependence and simply say polynomial time. LPs are quite fast in practice, outperforming their worst-case time complexity. As a check, references [$\alpha,\beta,\gamma$] which include “fair algorithms for clustering” do not include full time complexity guarantees. Further, the number of variables and constraints in the LP are polynomially sized. Specifically, we have $O(|\mathcal{C}|k)$ variables and $O(|\mathcal{C}|+|\mathcal{H}|+k|\mathcal{H}|)$ constraints.
>
> 8-$\textbf{Theorems cannot be verified experimentally}$: We are a bit unclear on the meaning of this statement and would appreciate clarification during the discussion phase.  Specifically, what does it mean that the statements cannot be verified in practice? All of the problems in fair clustering we are aware of have been NP-hard, and those prior works (as with ours) introduce approximation algorithms coupled, typically, with an experimental section. While no one obtains the optimal value by solving the NP-hard problem exactly, it follows from the fact that the approximation ratios are small, that the final output should not be large. Similarly, in our case one expects good performance in practice which is shown in Section 9.
>
> $\textbf{Limitations and Societal Impact}$: We believe our introduction section discussed the societal impact. Furthermore, our theoretical guarantees clarify the limitations of our algorithms. However, we can add a section to elaborate such issues.
>
> $\textbf{References}$:
>
> $\alpha$-Bera et al, “ Fair Algorithms for Clustering”, NeurIPS 2019.
>
> $\beta$-Esmaeili et al, "Probabilistic fair clustering", NeurIPS 2020.
>
> $\gamma$-Makarychev et al, "Approximation Algorithms for Socially Fair Clustering", arXiv preprint arXiv:2103.02512 (2021).

---

> > ### Comment · Reviewer_aWDm · 2021-08-11
> > **Whether the Problem is Meaningful and Experiments**
> >
> > Thanks very much for the author responses. Please let me further clarify my comments on **Whether the Problem is Meaningful** and **Experiments**. I am expecting more convincing evidence from the authors to demonstrate my current thought is not solid.
> >
> > **Whether the Problem is Meaningful** The extra explanation verified my previous understanding on the problem addressed in this paper was correct. Here I would like to provide my personal but maybe biased opinion on the general fairness clustering, which can be roughly divided into two categories.
> >
> > (a) Under the condition of meeting the fairness constraint, optimize the clustering cost. Most existing literature focuses on this category. But it suffers from one drawback. As the authors point out, the clustering cost is unbounded. I agree. Maybe no validate solution (empty solution) can be found if the fairness constraint is too harsh. For example, we require the portions of red and blue are exactly the same; however, the whole dataset is imbalanced in terms of red and blue.
> >
> > (b) With the upper bound of clustering cost, minimize the violation of fairness constraint. This paper is in this category. To my best knowledge, I did not see other papers in this category.
> >
> > Both of the above categories address fairness clustering. They are twins, different but related problems. Problem (b) criticizes (a) is unbounded, while (a) beats back by the fact that (a) does not meet all the fairness constraint. Therefore, I do not recognize the superiority of Problem (b). Instead, I, personally, more support Problem (a) with the following reasons. (1) The interior clustering performance from Problem (a) can be enhanced by relaxing the fairness constraint. Actually, the authors did relax the fairness constraint as well.  (2) Theoretically unbound does not mean that it does not work in practice.  For example, K-means with random initialization can be theoretically arbitrary worse. On the contrary, multiple runs with random initialization and picking up the minimum works well in practice. (3) For the concept PoF, it is difficult for practitioners to decide PoF = 1.1 or PoF = 1.2. Note that the output of cluster analysis is the partition, rather than clustering cost. Some clustering algorithms (AHC, DBSCAN) even do not have a cost. For some deep clustering algorithms, the practitioners do not care about the final objective function value. Instead, the violation of fairness constraint can be easily understood, which comes from the problem rather than the algorithm.
> >
> > Please kindly correct me if the authors found the above statement was wrong. Here I solicited the authors provide more evidence to convince me that Problem (b) is meaningful.
> >
> > **Experiments** This paper mainly focuses on the theoretical part and the experimental results in Section 9 are within expectation. Such experiments do not have too much value. Even I think it is not necessary. To make this paper more stronger, the authors can compare with the methods of Problem (a) in terms of both clustering cost and the violation of fairness constraint. Pareto curves are expected to demonstrate the effectiveness of the addressed problem and proposed method.

---

> > > ### Author Response · Authors · 2021-08-12
> > > **Response to: Whether the Problem is Meaningful and Experiments**
> > >
> > > We thank the reviewer for these careful followup questions and apologize for the lengthy response.
> > >
> > > $\textbf{Whether the Problem is Meaningful}$:
> > >
> > > Yes. Our work is in category (b) and to the best of our knowledge, it is the only work in (b) thus far (category (a) is more well-studied, but still a nascent area, only a few years old).
> > >
> > > We would like to be clear that we are not arguing that (b) is superior to (a), but rather that they serve different purposes and may be appropriate for different applications. Roughly speaking, if there is a hard fairness constraint that cannot be violated, then (a) is obviously more appropriate. On the other hand, if preserving some bound on a notion of cluster cost/quality is more important than strict fairness constraints, then (b) may be more appropriate. One motivating example we give for this is the legal concept that “disparate impact” (unintentional harm to some groups) can be allowed when it is tied to “business necessity”. In other words, some amount of unfairness can be allowed if it would be too costly to correct. An additional motivation for our problem definition along these lines is internal auditing. The structure of our problem is well-suited to engage with questions like, “How much fairer could we be if we allow a small increase in cost?”
> > >
> > > Another way to frame the difference is as follows. In (a), the practitioner must have the knowledge to choose appropriate fairness constraints (i.e. how far should the clusters deviate from demographic parity?), while in (b) they must have the knowledge to choose appropriate cost/quality constraints. Outside of scenarios with specific regulations or quotas on group fairness, we think it’s often reasonable to assume a user may have more knowledge about what is acceptable cluster degradation in their application area of expertise (b). The alternative provided by (a) is that they potentially choose an arbitrary fairness bound which “looks good”, but may lead to either poor clustering results or too weak fairness constraints.
> > >
> > > We now address the numbered concerns regarding how (a) and (b) compare:
> > >
> > > (1) It is correct that the clustering cost would be improved by relaxing the fairness constraints. But then the question becomes how should the relaxation be done? Which group should be relaxed, by how much, and what guarantees do we have? Our formulation provides answers by introducing the GROUP-UTILITARIAN and GROUP-EGALITARIAN objectives which have a strong grounding in welfare economics and by establishing approximation bounds on the objectives.
> > >
> > > (2) The comparison to k-means (we assume Lloyd’s algorithm) is not entirely accurate. In that case, a solution to the k-means objective exists. The heuristic algorithm is not guaranteed to find it, but works well at getting close enough in practice. However, we are saying that it can very easily be the case (even in practice) that satisfying given fairness constraints will degrade the clustering cost to the point that it is not useful or meaningful. This is due to the known issue of group membership being redundantly encoded in other features. Unlike with the k-means example given, no amount of randomized runs will be able to find a significantly better solution subject to the same fairness constraints. Further our experiments in section 9 show that we can achieve effectively zero proportional violations (“perfect fairness”) but only by letting the clustering cost increase.
> > >
> > > (3) First, we agree that there exist settings in which practitioners do not explicitly focus on the cost. However, we believe there are also settings in which they do. For example, if someone is clustering news articles daily, they may understand their distance metric well enough to know that a PoF of at most 1.2 will prevent two vastly different articles from being clustered together (e.g. it may change how articles are partitioned within the sports category, but won’t merge a sports cluster with a fashion cluster). The existence of clusters that are both meaningful and fair may change daily, but fixing the PoF at 1.2 can guarantee meaningful clusters while being as fair as possible for any given input.
> > >
> > > Second, the apparent simplicity of setting fairness constraints (a) can be an illusion. Since it is generally impossible to enforce perfect demographic parity, the challenging question for problem (a) is how much fairness constraints can/should be relaxed. While this may be obvious in some applications, it is a very difficult question in others.
> > >
> > >
> > > $\textbf{Experiments}$:
> > >
> > > We appreciate this clarification and feedback. We note that similar experiments are commonly used in prior work on other fair clustering problems where the results are within expectation. Further, we believe these experiments are expected by most reviewers and hence should be provided.

---

> > > > ### Comment · Reviewer_aWDm · 2021-08-12
> > > > **Whether the Problem is Meaningful**
> > > >
> > > > Thank the authors for the extra response. I am more convinced on the problem setting now and would like to increase my score. But how much the score will be increased will depend on the internal discussion among reviewers and AC.

---

### Official Review · Reviewer_9wxQ · 2021-07-15

**Rating:** 8
**Confidence:** 4

**Summary:**

The paper studies a new variant of fair clustering for different fairness (group utilitarian, group egalitarian, group leximen) and clustering objectives(k-center, k-median, k-means). The main approach is to optimize the fairness objective under the constraint that the clustering cost are bounded by some value U. The main result can be thought of as a bi-criteria approximation: If we allow that U is violated by a factor of (2+alpha) (where alpha is the approximation ratio of some clustering algorithm for the given objective) then we can get an additive approximation on the fairness objective.

In addition to this algorithmic upper bound, the authors also give a number of complexity theory lower bounds for different variants of the problem.

The authors also provide a proof-of-concept study of their algorithm.



**Limitations And Societal Impact:**

Yes.

**Main Review:**

The newly introduced variant of fair clustering is interesting and the proposed solutions are non-trivial. They provide a nice new approach to the fair clustering literature.

I would have loved to see some more details of the proof of Theorem 6.2 (maybe instead of some of the lower bounds). It seems that this is a central step in the paper and it would be helpful to have additional details in the main body of the paper. Also, how does this theorem relate to earlier fair clustering algorithms (for example, for 2 colors how does it relate to fairlets and similar ideas)?

In the experimental section there is no comparison to earlier work. I understand that the main results of the paper are the algorithmic/theoretical results and that the problem is somewhat different from the standard problem formulation, but I would still appreciate some comparision.

Overall, I like the paper and I recommend to accept it.

-----------------------------------------------------------------------
Thank you for your feedback. I will keep my score.




**Time Spent Reviewing:**

2

---

> ### Author Response · Authors · 2021-08-10
> **Response to Reviewer3(9wxQ)**
>
> $\textbf{Theorem 6.2 Issue}$: We agree with the reviewer that it is indeed an important step in the algorithm and essentially relates the problem to the assignment version where centers are fixed. Unfortunately, due to space limits we had to put it in the appendix. However, we would happily add a proof sketch for the sake of exposition as well. At a high level, the proof is constructive and the result is shown by considering the centers of the optimal solution (which are unknown), the centers of the color-blind clustering, and then following an assignment that routes points from an optimal center to its closest color-blind center. We then show that we establish bounds on the clustering cost and fairness objective by doing that. We believe this approach is simple and leads to many guarantees. It has been used before in literature (see [A,B,C]). If we were to compare it to the fairlet approach, then it is somewhat opposite. The fairlet approach essentially starts with the fairlets (which are like color-balanced “atoms”) and then they are clustered to yield a fair solution. On the other hand, here the centers are found and then we solve a fair (color-balancing) assignment “routing” problem. Therefore, the fairlet approach focuses on the color balance in the first step and ignores it in the second whereas this approach ignores the colors in the first step and balances them in the second. Although the fairlet approach does not involve LPs and is therefore more scalable, it does not seem to handle arbitrary lower and upper bounds (see the results of [D,E,F]).
>
>
>
>
>
>
>
> $\textbf{Issue with Experiments}$: We agree with the reviewer that adding baselines can certainly help. We plan to compare some baselines in mind such as exhaustive and heuristic search methods over the fairness parameters and we are open to suggestions during the coming discussion phase.
>
>
>
>
> $\textbf{References}$:
>
> A-Bera et al, “Fair Algorithms for Clustering”, NeurIPS 2019.
>
> B-Esmaeili et al, “Probabilistic Fair Clustering”, NeurIPS 2020.
>
> C-Brubach  et al. "Fairness, Semi-Supervised Learning, and More: A General Framework for Clustering with Stochastic Pairwise Constraints." AAAI(2021).
>
> D-Chierichetti et al, “Fair Clustering Through Fairlets”, NeurIPS 2017.
>
> E-Bercea et al,” On the Cost of Essentially Fair Clusterings”, APPROX/RANDOM 2019.
>
> F-Rösner et al, “Privacy preserving clustering with constraints”, ICALP 2018.

---

### Official Review · Reviewer_xNaF · 2021-07-16

**Rating:** 5
**Confidence:** 3

**Summary:**

EDIT: I increased my score following the discussions and rebuttal.
The main question in the paper is how can we cluster data so that we optimize for some fairness desiderata. In other words, if we have a target for the amount of clustering cost we are aiming for, how can we find the most fair solution, subject to that cost constraints? Put differently, the authors address the problem of minimizing the amount of "unfairness" subject to the clustering cost being below a certain threshold. The two desiderata put forth by the authors in terms of fairness have to do with "group utilitarian" objective and with "group egalitarian" (they also further study a generalization called "group leximin" objective. The authors give both positive results and some impossibility results. Finally, they run some experiments to validate their findings.

These three objectives are desiderata that have been studied in the past. Briefly, the group utilitarian objective minimizes the sum of all fairness violations, whereas the group egalitarian objective minimizes the maximum fairness violation. Going a step further, the leximin (LM) objective minimizes the maximum fairness violation first, then minimizes the second worst etc.

One of the hardness results they prove is that for all three objectives, both fairly clustering and fairly assigning points to fixed centers is NP-hard.

On the positive side, they give an approximation algorithm that yields a (2+a)U-approximation where U is an exogenous parameter that is the upper bound on the clustering cost and a is the approximation ratio of a color-blind clustering algorithm. The algorithms presented for the different objectives depend on some assumptions about the sizes of the clusters. The idea behind the algorithms is to first run a color blind clustering algorithm to select centers, followed by an LP to get a fair assignment and a flow computation.

The authors perform some computational experiments on the adult and the census data sets to evaluate how their fairness guarantees look like.


**Main Review:**

The reviewer believes the paper studies an interesting problem which has received a lot of attention recently by both theory and ML communities however the execution brought forth by the authors is done in a manner that has several caveats.

The reviewer overall is not convinced that the problem formulation itself yields interesting insights and it does feel a bit artificial. There are many prior works on fair clustering recently and to some extent the paper doesn't adequatly compare results: A paper here "Clustering without over-representation paper" (Ahmadian et al.) seems to be relevant and one natural question is, whether changing the representation parameters until a solution returned has value smaller than U would yield comparable results?  Could the authors provide experimental evidence to show their method significantly outperfoms prior methods?

Regarding other weaknesses, the paper has an exogenous parameter U for the bound in the clustering cost which is not clear how to get it in practice, and also the Assumption 6.1 makes their main result weaker than one would expect especially in the presence of outliers or misreported values. Can the Assumption 6.1 be removed or will the proof break down completely? What about the analysis of the LP then?

Overall, the theory here could be strong if the final theorems were cleaner but this is not the case. The paper has interesting directions and is currently a collection of small results but further work is needed to have a solid Neurips contribution.

**Time Spent Reviewing:**

5

---

> ### Author Response · Authors · 2021-08-10
> **Response to Reviewer2(xNaF)**
>
> 1-$\textbf{Comparing to Baselines}$: Doing a brute force search over an algorithm such as (Ahamdian et al) would require prohibitive time. For example, for the GROUP-UTILITARIAN objective even with two colors brute force is prohibitive (would have a complexity of $1 / \epsilon$), but our algorithm (see Theorem 6.5) speeds up the search to $\log(1 / \epsilon)$.
>
>
> 2-$\textbf{Assumptions (assumption 6.1)}$: The reviewer mentions that our algorithms depend on an assumption in the summary, and furthermore mentions Assumption 6.1 in the main review. We note that there is *no* such Assumption 6.1 in the paper and would appreciate the reviewer clarifying exactly which assumption is being referenced.
>
> Our algorithms are assumption free. A point brought up by R1(cQa8) is that the guarantees have a dependence on $L(U’)$ instead of $L(U)$ and that we may not have a guarantee over the bounds. This is true and we discuss this in Appendix C (specifically, Appendix C.2 and C.3). Furthermore, Theorem 5.3 shows algorithm-independent lower bounds that show that we are bounded by $L(U)$.

---

> > ### Comment · Reviewer_xNaF · 2021-08-13
> > **Appendix Assumption/Fact?**
> >
> > The reviewer thanks the authors for their careful response.
> >
> > The confusion comes as I was reading the full version of your paper with appendix and in page 32 you mention Assumption 5.1 multiple times actually (lines 1079-1084). Then once clicked, the Assumption 5.1 takes you to Fact 5.1 (line 177). Could you clarify if this is a Fact or an Assumption? If it's a fact and L(U) turns out to be relatively small, then can it happen that the guarantees of the main Theorem 5.3 become uninteresting? This is because all three parts of the theorem have L(U) in the denominator (e.g., the proportional violation of any color h is basically 1/L(U)). The reviewer would appreciate if the authors can help clarify these subtle points. Thank you!

---

> > > ### Author Response · Authors · 2021-08-13
> > > **Response to: Appendix Assumption/Fact?**
> > >
> > > We thank the reviewer for the followup questions and careful reading. We hope our response is not too lengthy.
> > >
> > > The word assumption in lines (1079-1084) is a typo. Section F.3 is essentially replicating section 9.3 for the additional experimental results we conduct in the appendix. This was a long submission (33 pages) and thus more prone to typos. We thank the reviewer for spotting the typo and will fix it.
> > >
> > > Regarding the guarantees of Theorem 5.3, we do not see how the results of Theorem 5.3 would be uninteresting in this case. Theorem 5.3 is not presenting guarantees on the performance of our algorithms, but rather intrinsic algorithm-independent lower bounds. Even another algorithm which could be superior would not overcome these lower bounds unless P=NP. We therefore think this shows that the problem has an inherent difficulty to it. In general, if we have an instance where $L(U)$ is indeed small, then Theorem 5.3 says that we we should not hope to produce an algorithm that gives a guarantee better than $\frac{1}{8L}$ for any color’s proportional violation since it is impossible unless P=NP. While it is a disappointing result, it is an inherent lower bound independent of the algorithm.
> > >
> > > Further, based on the bounds of Theorem 5.3, we have a more clear idea of the kind of problems where we can hope to achieve good performance. I.e., while a small $L(U)$ is an obstacle, a large $L(U)$ is not.
> > >
> > > Moreover, as discussed in our previous response above and also that with Reviewer1(cQa8) (beginning of the response), for the FCBC (clustering) problem our final guarantees are in terms of $L(U’)$ not $L(U)$, but for the FABC (assignment) problem our guarantees are in terms of $L(U)$ and are actually asymptotically optimal in terms of the lower bounds of Theorem 5.3 (see Theorem 6.7).
> > >
> > > Finally, we believe that such lower bounds that go beyond NP-hardness and give impossibilities over the approximations can guide future work on this problem.

---

> > > > ### Author Response · Authors · 2021-08-21
> > > > **Follow up**
> > > >
> > > > We thank the reviewer for the above feedback and we were wondering if they have any further comments/questions about our response. The rest of the reviewers have so far given us positive reviews, so please do not hesitate to ask for any further clarifications.

---

### Official Review · Reviewer_cQa8 · 2021-07-18

**Rating:** 6
**Confidence:** 3

**Summary:**

The paper studies the fair clustering (with balanced clusters requirement) and the goal of authors is to achieve the “fairest” possible clustering achievable with cost at most U. The idea here is that we know that PoF can be unbounded, and it is unrealistic to assume that in practical applications we cannot sacrifice the clustering quality for fairness with no limit.
An important component is how to measure the amount of unfairness and then the goal is to minimize it while not violating the upper bound on affordable clustering cost. They have defined three notions called GROUP-UTILITARIAN, GROUP-EGALITARIAN and GROUP-LEXIMIN. Basically, in fair clustering, for each color h, we are given lower and upper bounds $\alpha_h, \beta_h$ and the goal is to guarantee that in each cluster the number of point from color h is at least $\alpha_h \cdot  |C_i|$ and is at most $\beta_h \cdot |C_i|$. Now, the amount of unfairness is defined by a new set of parameters $\Delta_h$ such that instead of satisfying the fair clustering requirements exactly, we have that the number of points from color h is at least $(\alpha_h - \Delta_h) \cdot  |C_i|$ and is at most $(\beta_h + \Delta_h) \cdot |C_i|$. The authors consider $\min \max_h \Delta_h$ and $\min sum_h \Delta_h$ objectives (GROUP-LEXIMIN is a lexicographical minimizing of \Delta_h: the top priority is to minimize the min value, then among those with the same min value, minimize the second min value and so on).


**Limitations And Societal Impact:**

The paper works on theoretical foundation of fairness and I do not see any immediate potential negative societal impact of their work.

**Main Review:**

The theoretical guarantees of the paper are for both fair clustering with bounded cost and the fair assignment under bounded cost (i.e., a set of centers is given, and the goal is to find the best possible fair assignment under bounded cost). Both problems are NP-hard. Further they showed that the violation (which is a function of $\Delta_h$) cannot be set to arbitrarily small values (the values depend on the size of smallest non-empty cluster across all clusterings of bounded cost); otherwise, the problem becomes NP-hard (the actual bounds are given in Thm 5.3).
On the algorithmic side, they provide a (semi-)bicriteria approximation for fair clustering under bounded cost: the algorithm finds a solution of cost at most $(2+ \gamma) OPT$ (where $\gamma$ is the approximation factor for the used color-blind clustering algorithm) with almost additive approximation optimal w.r.t. $(2+ \gamma) OPT$ and the bounds given in Thm 5.3. The approach is via a reduction from the fair algorithms under bounded cost.

**Questions for Authors**
1)	Do you have any evidence showing the bound of Thm 6.1 is reasonable? L(U’) can be arbitrarily smaller than L(U). Do you have any evidence that such dependence is required? I would like to get a sense how tights your approach is.

**Minor Comments**
1.	In abstract: “price of fairness,” -> “price of fairness”,
2.	Theorem 6.1 is not mentioning the exponential dependence on \eps in the running time. Alternatively, you can mention the number of colors is a constant.
3.	Line 198. Providing 159 without mentioning the other parameter in the dataset (e.g., number of points, centers) may not be very helpful.
4.	Line 203. A space is needed after GROUP-UTILITARIAN,
5.	Line 225. Netwok -> network
6.	Line 340 and 341 does not seem very informative. Also reference [8] does not provide any such evidence for fair clustering. Maybe you meant the result of [10]?
7.	In section 9.3, it is not clear how you are computing L(U)? Ho do you consider all possible clustering of certain bound (even approximately)? Would be helpful to elaborate more here.
8.	In the second line of the equation after line 732, \phi(i) -> \phi(j)
9.	Rephrase line 735
10.	Line 750. Isn’t the proof of Theorem 6.3 trivial? Given that we need to satisfy both upper and lower bound? In fact, why is it important to know the non-convexity for the case we are ignoring the upper bounds?
11.	Line 822. theorem Theorem -> Theorem
12.	Line 905. Rephrase and fix the typo
13.	Line 911. We havewe no -> we have no

The problem statement and the techniques used in the paper are interesting. The drawback is that it is not clear why the result cannot be (easily) improved. Authors’ feedback on this point can be very helpful for the decision process.

=====================================================

I would like to thank the authors for their helpful response. I will keep my score for this paper.

**Time Spent Reviewing:**

3

---

> ### Author Response · Authors · 2021-08-10
> **Response to Reviewer1(cQa8)**
>
> $\textbf{Our Response to the main question about Theorem 6.1}$: We had indeed made similar observations (please see appendix C, especially C.2 and C.3, in the supplementary material). The fact that there is no bound on $L(U’)$ in terms of $L(U)$ is a shortcoming. However, if the example is not pathological and the clusters exhibit “good” separation then we do not expect the difference to be very large.
>
> Additionally, for the (assignment) FABC problem (i.e., the case where centers are already chosen) our guarantees are actually in terms of $L(U)$ (Theorem 6.7) and therefore asymptotically match the lower bounds (Theorem 5.3).
>
> We have attempted to make the approximation be in terms of $L(U)$ instead of $L(U’)$, the issue lies in the fact that we could not approximate the fairness objective without a violation to the clustering cost. We believe that such issues could be addressed in future work.
>
>
> $\textbf{Our Response to the minor points}$:
>
> We will fix the mentioned typos, here we address the more complicated points in their order:
>
> 2-Theorem 6.1: Indeed for many applications the number of colors is constant and thus the algorithm would run in polynomial time. So we will add a note about that. It is also worth noting that this step can be parallelized.
>
>
> 3-Line 198: The number of points used for a given dataset is mentioned in the beginning in Section 9. The minimum number of points (e.g., 159) is the minimum across all choices of values for the number of centers $k$ and PoF. We believed that providing the minimum (worst case) across the choices of $k$ and PoF would be best. As the reviewer mentions, perhaps something like a plot of the minimum size vs PoF for a given choice of $k$ would also be useful; we are happy to add this to the paper. In fact here is a partial table for the adult dataset for $k=15$:
>
> | PoF  | Minimum Cluster Size |
> |---------|----------------|
> |1.001       |        159 |
> |1.005       |       164  |
> |1.011       |       168  |
> |1.020       |       175 |
> |1.026       |       179 |
> |1.03        |       180 |
>
>
>
>
>
>
> 6-Line 340 and 341: For arbitrary lower and upper proportion bounds in an arbitrary metric space, a true approximation algorithm has not been introduced, where a true approximation algorithm is one that only approximates the objective but does not violate the constraint (this is noted in the footnote in the bottom of page 8). While it is true that [10] provides a bi-criteria (not a true approximation) algorithm, references [13] and [8] explicitly mention the lack of true fair clustering approximation algorithms in the literature.
>
>
>
>
> 7-Not clear how to compute $L(U)$: As mentioned in lines 403 and 404, it is intractable to find the values of $L(U)$. By definition, $L(U)$ is the size of the smallest cluster across all solutions of cost not exceeding $U$. A larger $L(U)$ corresponds to better approximation (Theorems 5.3, 6.1, and 6.7). Therefore, $L(U)$ provides us with the worst-case additive approximation. If we are able to find an LP solution of cost $U$ and size larger than $L(U)$, then rounding it would introduce a smaller violation, than rounding an LP solution of size $L(U)$. Therefore, our point in that section was to check what the value of the smallest cluster is; had it been small, say 4, then that would mean that the size of the smallest cluster is posing a significant obstacle to getting good approximations. However, we see that we actually achieve clusterings of size at least 159.
>
> It might seem reasonable to consider maximizing the size of the largest cluster to get smaller violations. This would work in practice, however theoretically (in the worst-case) we can have examples where all clustering of cost not exceeding $U$ have the same size.
>
>
>
>
> 10-Proof of Theorem 6.3: There are many possible settings in fair clustering that have been considered in the literature. For example (Chierichetti et al, 2018) only considers lower bounding the proportions, whereas (Ahmadian et al, 2019) had considered only upper bounding the proportions. We therefore considered only the lower bound case to show that the set is indeed non-convex even if we were to “simplify” the problem and only consider the lower proportion bounds. We note at the end of the proof (lines 768-769) that through very similar methods we could also show a similar result had we only considered the upper bounds. It follows that when both bounds are considered that the set is non-convex. The non-convexity of the set prohibits the direct application of methods from convex optimization which we think is an important point to make.
>
>
> $\textbf{Question on why the results cannot be easily improved}$:
> We believe that the solution we have presented works in practice and can in fact solve multiple clustering cost functions. Moreover, we obtain fundamental algorithm-independent lower bounds on the problem (see Theorem 5.3) which we think are important and sometimes absent in the fair clustering literature. One source of difficulty is the non-convexity of the set (Theorem 6.3), which makes it difficult to apply convex relaxation methods. We do suspect that there are possible hardness results that make the problem generally difficult even using other methods.
>
> Moreover, since in this problem we are minimizing the proportional violations ($\Delta_h$) which are arbitrary fractional numbers in [0,1], the approximation guarantees that we have to obtain become more stringent. For example, constant multiplicative factors (which are not difficult to obtain) are meaningless (in the worst case), so are constant additive approximations. The only valid approximations are Polynomial-time approximation scheme (PTAS) guarantees where we can get a solution that is epsilon away from the optimal for an arbitrarily small epsilon.

---

### Author Response · Authors · 2021-08-10
**Reply to All Reviewers**

We thank the reviewers for their careful reading and constructive criticism. Please, see below each review for the corresponding response.

---

### Decision · Program_Chairs · 2021-09-27

**Decision:**

Accept (Poster)

**Comment:**

The paper proposes new formulations of (and algorithms for) fair clustering. Instead of minimizing clustering cost under fairness constraints, the paper considers a dual problem where the clustering cost is given and the goal is to maximize fairness.

The reviewers found the formulation of fairness to be interesting and potentially leading to a follow-up line of research. Almost all reviewers recommended acceptance, sometimes strongly so.  At the same time, there was a concern that the approximation factors depend inversely on the cluster size, so they could be large. However, overall, the reviewers felt that the positives outweigh the negatives.